# Stable and low-precision training for large-scale vision-language models

**Mitchell Wortsman**[*1]  **Tim Dettmers**[*1]  **Luke Zettlemoyer**[12]  **Ari Morcos**[†2]

**Ali Farhadi**[†1]  **Ludwig Schmidt**[†134]

## Abstract

We introduce new methods for 1) accelerating and 2) stabilizing training for large language-vision models. 1) For acceleration, we introduce SwitchBack, a linear layer for int8 quantized training which provides a speed-up of 13-25% while matching the performance of bfloat16 training within 0.1 percentage points for the 1B parameter CLIP ViT-Huge—the largest int8 training to date. Our main focus is int8 as GPU support for float8 is rare, though we also analyze float8 training through simulation. While SwitchBack proves effective for float8, we show that standard techniques are also successful if the network is trained and initialized so that large feature magnitudes are discouraged, which we accomplish via layer-scale initialized with zeros. 2) For stability, we analyze loss spikes and find they consistently occur 1-8 iterations after the squared gradients become under-estimated by their AdamW second moment estimator. As a result, we recommend an AdamW-Adafactor hybrid which avoids loss spikes when training a CLIP ViT-Huge model and outperforms gradient clipping at the scales we test.

## 1 Introduction

Large models trained on large datasets have recently led to multiple breakthroughs in machine learning such as GPT-3 [5] and PaLM [11]. While many components are necessary for successful large-scale training, two critical elements are training speed and stability. To enable further progress, we must ensure that 1) training is fast—the model should be able to see a lot of data even if it is large, and 2) training is stable—large models should not suffer from loss spikes which degrade performance. We study these two directions in the context of contrastive language-image pre-training (CLIP) [46]. We examine CLIP-style models because of their importance in computer vision: CLIP-style models reach state-of-the-art performance on a wide range of image classification tasks [46, 64, 44, 7] and underlie image generation methods such as DALLE·2 [49] and Stable Diffusion [51]. Our contributions towards fast training and stable training are as follows.

**Towards fast training**, we introduce **SwitchBack**, a linear layer for quantized training with int8 precision which matches the performance of the bfloat16 [63] baseline within 0.1 percentage points for CLIP ViT-Huge—a larger model than considered in the original CLIP paper. Linear layers account for the majority of the compute in standard transformer models, usually more than 90%, comprising the key, query, value, and out projection of the attention blocks as well as the multilayer perceptron. We perform all linear layers in low-precision (int8) while retaining other layers, such as layer norms, in higher precision. With this setup, we observe end-to-end speedups between 13 and 25% for CLIP ViT-Huge training: 25% compared to a standard linear layer implemented using the PyTorch [43] autograd python module and 13% compared to the standard PyTorch layer which include background CUDA/C++ optimizations which are difficult to replicate for custom layers.

---

[1]University of Washington. [2]Meta AI Research, FAIR Team. [3]Allen Institute for AI. [4]LAION. [*]Equal contribution. [†]Equal senior contribution.

37th Conference on Neural Information Processing Systems (NeurIPS 2023).

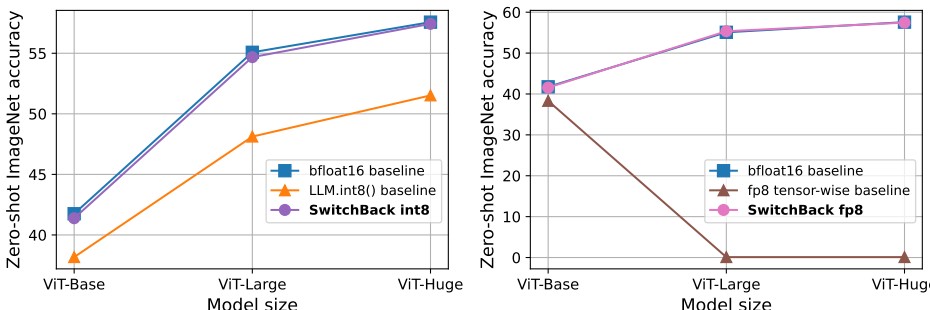

Figure 1: We introduce SwitchBack, a linear layer for low-precision training. **(Left)** SwitchBack for int8 training matches the zero-shot ImageNet [15] accuracy of standard bfloat16 training within 0.1 percentage point for CLIP ViT-Huge [46, 20] and outperforms LLM.int8() [17]. **(Right)** For float8 (fp8) training [40], a baseline which uses tensor-wise quantization diverges for large models while SwitchBack matches the baseline. In these large-model, small-data experiments, our focus is on comparing methods and not final model accuracy, so we use short runs which makes it feasible to run many experiments.

SwitchBack starts from the observation that quantization noise grows with the inner dimension in a matrix multiplication. For CLIP training, the weight gradient computation involves a large inner dimension because CLIP training requires a large batch size [44]. Hence SwitchBack uses 16 bit precision matrix multiplication for the weight gradient computation while using int8 multiplications for the forward pass and layer input gradient computations. This approach leads to large accuracy improvements compared to LLM.int8() [17] (Figure 1). We will provide open-source Triton [57] kernels for Switchback to enable future work on efficient quantization schemes.

Besides int8 training, we also study large-scale 8-bit float (fp8) [40] training. We do not have access to hardware that supports fp8 data types, which is currently more rare than int8, so we use an accurate simulation of fp8 computation. SwitchBack also outperforms straightforward 8-bit float (fp8) baselines because tensor-wise quantized baselines diverge at >420M scale (Figure 1). However, we demonstrate that these methods can achieve high accuracy if the network is trained while keeping feature magnitudes small, which we accomplish via layer-scale [58] initialized with zeros.

**Towards stable training**, we find that loss spikes occur in CLIP training when the AdamW [37] second moment estimator becomes out-of-date in the patch embedding [20] layer. In particular, the learning signal changes so that the moving averages of squared gradients underestimates their true magnitude. Indeed, in the absence of stability interventions, we show that loss spikes can be predicted by examining this ratio of the squared gradients to their moving average. We therefore recommend an AdamW-AdaFactor [54] hybrid, which we refer to as **StableAdamW** as it removes instabilities at the scales we consider and outperforms gradient clipping. Concretely, StableAdamW is AdamW with the update clipping technique introduced in AdaFactor. Update clipping tracks the average ratio of the gradient square to the second moment estimator and lowers the learning rate when the ratio is large.

The remainder of this paper is organized as follows: Section 2 focuses on low-precision training while Section 3 stabilizes training by reducing loss spikes.

## 2    8-bit training

This section develops and compares methods for eight-bit training of languge-vision transformer models. First, Section 2.1 discusses preliminaries and related work. Next, Section 2.2 introduces and tests SwitchBack, a linear layer for int8 and float8 training. Finally, Section 2.3 develops alternatives to SwitchBack which can be used for float8.

### 2.1    Preliminaries and related work

Neural networks today typically use 16-bit operations for training [39] in either the float16 or bfloat16 format [63]. Floating point formats use a subset of bits to represent the exponent while the remainder specifies the fraction (often referred to as the mantissa). The float16 format uses 5 bits for the exponent while bfloat16 uses 8 and therefore covers a larger range—float16 has a range of $(5.96 \cdot 10^{-8}, 65504)$ while bfloat16 has a range of $(10^{-38}, 3 \cdot 10^{38})$. Most floating point formats also have denormalized numbers which allow for a "soft underflow" which gets exponentially closer to 0.0f for each additional

Algorithm 1: PyTorch code for **SwitchBack**

```
class SwitchBackMatmul(autograd.Function):
    @staticmethod
    def forward(ctx, X, W):
        ctx.save_for_backward = X, W

        X_int8, state_X = row-wise_quantize(X)
        W_int8, state_W = tensor-wise_quantize(W)

        return matmul_int8_and_dequanitze(
            X_int8, W_int8.t(), state_X, state_W
        )

    @staticmethod
    def backward(ctx, G):
        X, W = ctx.save_for_backward

        G_rowwise = rowwise_quantize(G)
        W_int8, state_W = tensor-
            wise_quantize_transpose(W)

        X_gradient = matmul_int8_and_dequanitze(
            G_int8, W_int8.t(), state_X, state_W
        )
        W_gradient = matmul_fp16(G.t(), X)

        return X_gradient, W_gradient
```

Algorithm 2: StableAdamW $(\{\alpha_t\}, \beta_1, \beta_2, \epsilon)$

$v_0, u_0 = \mathbf{0}$
**for** $t = 1$ **to** $T$ **do**
$\quad g_t = \nabla f(\theta_t)$
$\quad$ // apply correction term to debias EMA.
$\quad \hat{\beta}_1 = \beta_1 \cdot \frac{1-\beta_1^{t-1}}{1-\beta_1^t}$
$\quad \hat{\beta}_2 = \beta_2 \cdot \frac{1-\beta_2^{t-1}}{1-\beta_2^t}$
$\quad$ // update moving averages
$\quad v_t = \hat{\beta}_1 v_{t-1} + (1 - \hat{\beta}_1)g_t$
$\quad u_t = \hat{\beta}_2 u_{t-1} + (1 - \hat{\beta}_2)g_t^2$
$\quad$ // for implementation convenience
$\quad$ // operations below occur independently
for each tensor
$\quad \mathsf{RMS}_t = \sqrt{\mathbb{E}\left[g_t^2/u_t\right]}$
$\quad$ // update parameters
$\quad \eta_t = \alpha_t/\max\left(1, \mathsf{RMS}_t\right)$
$\quad \theta_t = \theta_{t-1} - \eta_t\lambda\theta_{t-1} - \eta_t v_t/\left(\sqrt{u_t} + \epsilon\right)$

bit in the mantissa. To prevent underflows float16 mixed precision training [39] has been developed which works as follows. The loss of a mini-batch is multiplied by a loss scalar to scale the loss and following backpropagation gradients into the representable range of fp16. This loss scaling is undone by rescaling the weight gradients before the optimizer updates fp32 main weights with the fp16 gradients. In PyTorch [43], the loss scalar is initialized to 65536. Everytime an Inf/NaN is encountered, the update is skipped and the loss scalar is halved. If no Inf/NaN are encountered for 2k iterations, the scalar is doubled.

When the loss scalar becomes too low in float16 training the loss slowly diverges. This was observed by Cherti et al. [9] when training ViT-Huge CLIP models and remedied by switching to bfloat16. Another instance of float16 creating issues at scale was the training of OPT [73] and BLOOM models [52]. Indeed, many obstacles faced during the OPT project could have been alleviated by using bfloat16 [72]. Similarly, all float16 training runs for BLOOM ended in divergence, only after using bfloat16 was the training stable. However, fast bfloat16 support is only available on TPUs, or GPUs developed with or after the NVIDIA Ampere series (2021 or later).

While 16 bit training is the standard today, hardware support for 8 bit operations are becoming more common. Hopper GPUs support float8 (fp8) [40] and Ampere GPUs support int8. However, it is currently (2023) very difficult to attain Hopper GPUs. Moreover, while int8 and int4 are used for inference [17, 65, 16], and there is earlier work exploring 8 bit training for convnets [61, 78, 10], these formats are not commonly used for training transformer models at scale. The CLIP ViT-Huge models we train have 1B parameters including the image and text towers which is 40x larger than a standard ResNet-50 (23M) [28], and quantization is more challenging for large tensors [17]. Additional related work on quantization of large scale models (larger than BERT-large) and low-precision training and be found in Appendix A.

## 2.2 SwitchBack

### 2.2.1 Method

**Overview.** A linear layer consists of three matrix multiplications—one in the forward pass to compute outputs and two in the backwards pass to compute gradients for the input and weights. Our SwitchBack layer uses 8 bit precision for the first two matrix multiplies but switches back to higher precision for the weight gradient.

We compute the weight gradient in higher precision because this matrix multiplication involves dot products between vectors which have a length of batch size times sequence length. As CLIP training requires large batch sizes [46, 44], this inner dimension of batch size times sequence length is much larger than for the other matrix multiplies. As we show in Appendix D, variance due to quantization

increases with the inner dimension of the matrix multiply. This modification is what differentiates SwitchBack from LLM.int8(), allowing SwitchBack to match the bfloat16 baseline (Figure 1).

**Notation.** A standard linear layer is comprised of inputs $X \in \mathbb{R}^{b \times n}$, weights $W \in \mathbb{R}^{m \times n}$, and outputs $Y \in \mathbb{R}^{b \times m}$. In the forward pass, outputs are computed as $Y = XW^\top$. In the backwards pass the layer receives gradients of the loss with respect to $Y$, which we denote $\dot{Y}$. Then, gradients to inputs $\dot{X}$ are computed via $\dot{X} = \dot{Y}W$ while gradients to the weights $\dot{W}$ are computed via $\dot{W} = \dot{Y}^\top X$. For linear layers in a transformer [60], $b$ is batch size times sequence length, while $n$ and $m$ are small multiples of the embedding dimension.

**Quantization.** For the matrix multiplies in 8 bit precision we use quantization. There are a multiple quantization techniques to choose from and we will release code for all these alternatives. However, we find the best trade-off of simplicity and performance is from using i) row-wise quantization [31] for the inputs and gradients and ii) tensor-wise quantization for the weights. Additional information on quantization methods is provided by Dettmers et al. [17] but we summarize below. Using int8 as an example, which can represent integers from $-127$ to $127$, we now define row-wise and tensor wise quantization. For a matrix $X$ with rows $x_1, ..., x_b$, row-wise quantization $Q_{\text{row}}$ and tensor-wise quantization $Q_{\text{tensor}}$ are given respectively by

$$Q_{\text{row}}\left(\begin{bmatrix} x_1 \\ \vdots \\ x_n \end{bmatrix}\right) = \text{round}\left(\begin{bmatrix} \frac{127}{\text{absmax}(x_1)} \cdot x_1 \\ \vdots \\ \frac{127}{\text{absmax}(x_b)} \cdot x_b \end{bmatrix}\right), \quad Q_{\text{tensor}}(X) = \text{round}\left(\frac{127}{\text{absmax}(X)} \cdot X\right) \quad (1)$$

where $\text{absmax}$ is the maximum of the absolute value.

Importantly, when applying $Q_{\text{row}}$ we also save the row-wise absolute maximums so that we can use them later for dequantization. We refer to this as the quantization state, or *state*, for short, so $\text{state}_{\text{row}}(X) = [\text{absmax}(x_1), ..., \text{absmax}(x_b)]^\top \in \mathbb{R}^{b \times 1}$. Equivalently, for tensor-wise quantization we only need to store the tensor-wise absolute maximum so $\text{state}_{\text{tensor}}(X) = \text{absmax}(X) \in \mathbb{R}$.

Since only the matrix multiply occurs in int8 precision we need to dequantize the outputs back to the original floating point precision. The forward pass with quantization and dequantization becomes

$$\frac{\text{state}_{\text{tensor}}(W)}{127^2} \cdot \text{state}_{\text{row}}(X) * \underbrace{Q_{\text{row}}(X) \, Q_{\text{tensor}}(W)^\top}_{\text{int8 matmul}} \quad (2)$$

where $*$ denotes elementwise-multiplication, which in this case is broadcasted so that row $i$ of the matrix $Q_{\text{row}}(X) \, Q_{\text{tensor}}(W)^\top$ is multiplied by element $i$ of $\text{state}_{\text{row}}(X)$.

As mentioned previously, we use row-wise quantization for the inputs and gradients and tensor-wise quantization for the weights. We find that using row-wise quantization for both matrices increases complexity at a negligible or no performance increase. As such, we use this simpler approach.

The last detail in our algorithm is hardware specific. NVIDIA GPUs, which we use in this work, do not implement the int8/float8 operation $AB$ for matrices $A$ and $B$ and only $AB^T$ is implemented. As such, it is necessary to transpose the weight matrix in the backward pass. To reduce the overhead of transposition and quantization we fuse both operations, meaning we load the required data once from slow DRAM into fast SRAM/shared memory and then perform both operation in this cached memory – this is critical for achieving speedups. We call this operation `tensor-wise_quantize_transpose`, which is a fused tensor-wise quantize and transpose operation. Putting the pieces together, the result is Algorithm 1.

**Variants.** While Algorithm 1 is the most straightforward version of SwitchBack, we also present two alternative versions—SwitchBackM and SwitchBackQ—and will release triton [57] implementations for all three. Appendix C contains pseudocode. SwitchBackM (Algorithm 3) is a memory efficient version of SwitchBack which only saves 8 bit tensors for the backwards pass—we recommend its use when memory is limited. The small downside of SwitchBackM is that it requires an additional dequantize operation during the backwards pass which increases the runtime overhead. For CLIP ViT-Huge we observed only a negligible accuracy differences between SwitchBack and SwitchBackM. In addition, we present SwitchBackQ (Algorithm 4) which uses row-wise and column-wise quantization for the weights instead of tensor-wise. While we did not observe this to improve accuracy at the scales we consider, it's possible that it will perform better than SwitchBack at larger scale.

**float8.** While the explanation so far has used int8 as an example, the code for SwitchBack and float8 (fp8) is nearly identical. The only modification is that operations such as $\mathsf{round}(127x/\mathsf{absmax}(x))$ are replaced by $\mathsf{float8cast}(x/\mathsf{absmax}(x))$ where we simulate $\mathsf{float8cast}$ by rounding to the exact values of the float8 data type. This simulation improves on the simulation of [40] which only clips the input tensors into the representable range of the float8 data type, but not the exact values of the float8 data type. This simulation theoretically matches float8 training, but we are unable to perform real float8 training because we lack the hardware that supports float8 arithmetic. As such, we perform arithmetic in 16-bit with exact float8 values. For our int8 experiments we conduct the multiplications in int8 using A100 GPUs—we perform real int8 training without any simulation.

### 2.2.2 Experimental setup

To evaluate SwitchBack we train CLIP [46] visual transformer [20] models on LAION-2B [53]. Typically CLIP training, especially at ViT-Huge scale, is prohibitively expensive. Our goal is not high final accuracy but rather to contrast different methods for low-precision training. To enable running multiple experiments, we therefore only train for a small number of samples seen—380 million images—and use patch-dropout 0.5 [35]. We note that the experiment is still very expensive, corresponding to roughly 300 epochs of ImageNet training in terms of samples seen, or approximately 2.9e20 FLOPs per training run. After training on LAION-2B we evaluate the models zero-shot on ImageNet [15] using the 80 prompt templates from CLIP [46].

We use batch size 16384 (per-gpu batch size of 256) and train for a total of 20k iterations. The first 5k iterations are linear warmup while the remaining 15k are cosine decay. Training and evaluation are conducted with the OpenCLIP library [29] with learning rate 2e-3, weight decay 0.2, and batch-size 16384 using the optimizer described in Section 3.5.

### 2.2.3 Results

We test two main questions: (1) can we replicate 16-bit performance with SwitchBack and (2) can we get speedups. To test (1) we train CLIP models with SwitchBack across multiple scales with both int8 and float8 precision (Figure 1). To test (2) we profile operations in an individual linear layer and also measure end-to-end training speed. Loss curves for the training runs in Figure 1 are shown in Appendix Figure 9.

**Accuracy.** We find that SwitchBack can match standard 16-bit training performance and outperform baselines for both a) int8 precision and b) float8 precision.

For our int8 experiments (Figure 1, right), we contrast the performance of i) the standard baseline which uses mixed-precision bfloat16, ii) the matrix multiplication kernels from LLM.int8() [17], which is equivalent to SwitchBackQ (Algorithm 4) if the weight gradient multiplication was also performed in int8 using row- and column-wise quantization, and iii) SwitchBack. SwitchBack has a negligible accuracy drop of 0.1 percentage points compared to the bfloat16 baseline for CLIP ViT-Huge. In contrast, there is a drop of 5.9 percentage points when training with LLM.int8(). Section D details our hypothesis for why LLM.int8() fails to replicate 16-bit performance for CLIP training.

For our simulated float8 training experiments (Figure 1, right), we contrast the performance of i) the standard baseline which uses mixed-precision bfloat16, ii) a baseline which uses tensor-wise quantization for all matrices, that is the weights, inputs, and gradients, and iii) SwitchBack. SwitchBack has a negligible accuracy drop of 0.1 percentage points from the bfloat16 baseline for CLIP ViT-Huge. In contrast, training diverges for the baseline that uses tensor-wise quantization for all matrices.

**Speed.** By writing custom triton kernels [57] we achieve end-to-end speedups from 13-25% for CLIP ViT-Huge training. 25% compared to a standard linear layer implemented using the PyTorch [43] autograd python module and 13% compared to the standard PyTorch layer which include optimizations that are difficult to replicate for custom layers. Moreover, we find that the overhead due to quantization operations decreases with scale and is ∼10% for CLIP ViT-Huge. A detailed analysis of our speed-ups is in Appendix B.

### 2.3 Float8 training by reducing feature magnitude

We find that SwitchBack is necessary for high accuracy int8 training. However, this section develops other interventions which enable float8 training without SwitchBack. We show that high accuracy can be achieved via float8 training with tensor-wise quantization for the inputs, weights, and gradients, so

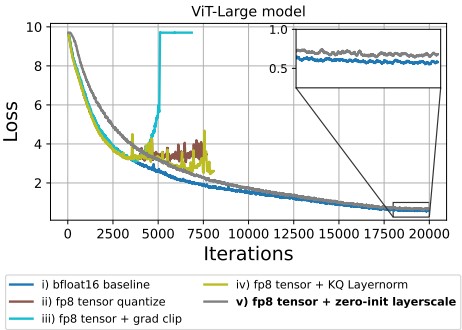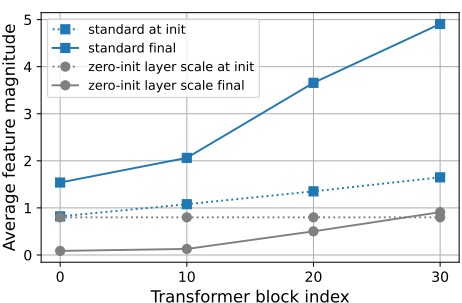

Figure 2: **(Left)** Training CLIP ViT-Large models with simulated fp8 precision using tensor-wise quantization for the inputs, weights, and gradients. All methods we try diverge except for using *zero-init layerscale* [58], which multiplies the output of each self-attention or mlp block with a learnable vector initialized to zero. **(Right)** Examining feature magnitudes (i.e., the average absolute value of the output for transformer block $k$) for CLIP ViT-Huge at the beginning (init) and end of training. This suggest why zero-init layer scale enables float8 training—zero-init layer scale prevents high feature magnitudes which may cause issues for low precision training [17]. Without the intervention, the average feature magnitude becomes large for later blocks.

long as the network is initialized and trained in a way which discourages large feature magnitudes. We accomplish via layer-scale [58] initialized to zero.

We use the bitsandbytes library [18] to simulate float8 training using the fp8 types from Micikevicius et al. [40]. We use tensor-wise quantization for the inputs, weights, and gradients, so that all operations occur in simulated float8. In our simulation, we represent each value only with the exact values representable by float8, but we perform computations in float16 precision. We believe that tensor-wise quantization approximates the removal of quantize operations entirely. This is because, as we show in Appendix C.2 (Figure 11), the maximum of these tensors tends to evolve smoothly. Consequently, using a moving average for a maximum which is divided directly in the matmul is similar to tensor-wise quantization.

Layer-scale, introduced by Touvron et al. [58], scales each self-attention and MLP block output hidden state by a learnable vector of shape embed_dim. A pre-norm transformer block with layer-scale tensors $\gamma_1$ and $\gamma_2$ is defined as

$$x'_k = x_k + \gamma_1 * \mathsf{self\_attention}(\mathsf{norm}_1(x_k)), \quad x_{k+1} = x'_k + \gamma_2 * \mathsf{mlp}(\mathsf{norm}_2(x'_k)), \qquad (3)$$

where $*$ is broadcasted elementwise multiplication.

Typically, layers are initialized so that they approximately preserve the variance of their inputs, and inputs have approximately unit variance [26, 27]. However, when combined with residual connections this can lead to higher norms in deeper networks.

Consequently, researchers have proposed initialization and scaling schemes which remedy this issue [1, 71, 4, 19]. Layer-scale with initialization 0 is an example of one such scheme—at initialization the transformer is an identity function. While $\gamma_1, \gamma_2$ are typically initialized as vectors of $10^{-4}$ or $10^{-6}$, we use 0 for simplicity.

Figure 2 (right) demonstrates that the layer-scale intervention is successful at controlling the average magnitude output. Without the intervention, the average feature magnitude $\mathbb{E}[\mathsf{abs}(x_k)]$ becomes high for later blocks. Previous work [17] has shown that large feature magnitudes result in issues for low precision training.

Results for simulated fp8 training are shown in Figure 2 (left) for ViT-Large. We find that all fp8 runs diverge except for when we use layer-scale initialized to zero. Concretely, Figure 2 compares i) the baseline which uses bfloat16 training, ii) using fp8 with tensor-wise quantization and no further modifications, which slowly diverges, iii) adding gradient clipping to ii), which also diverges, iv) adding KQ layernorm [14] to ii), which also diverges, and v) using *zero-init layerscale*, which trains without diverging. While there is a difference still between fp8 and bfloat16 training, this is primarily because of layerscale. Moreover, we believe that with hyperparameter tuning layerscale would match standard training in terms of accuracy.

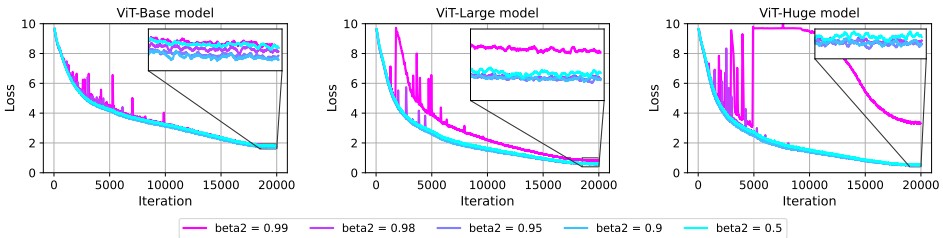

Figure 3: Loss spikes increase with **model size** for fixed learning rate and batch size. Reducing AdamW $\beta_2$ from its default in PyTorch of 0.999 mitigates loss spikes. Reducing $\beta_2$ too much slows training.

# 3 Stability

We now switch focus from accelerating learning by reducing precision to addressing instabilities which can arise during training. Section 3.1 reviews preliminaries and related work while Section 3.2 details the experimental setup. Next, Section 3.3 examines trends for training instability, finding loss spikes to increase with model scale but decrease with lower AdamW $\beta_2$. Then, Section 3.4 finds that loss spikes arise in our setting due to an out-of-date AdamW second moment estimator leading Section 3.5 to adopt and tests a fix developed in the context of AdaFactor [54]. Appendix Section G connects this section on stability with the previous section on low-precision training.

## 3.1 Preliminaries and related work

Loss spikes can emerge when scaling up models [8, 25, 14, 68, 70, 54, 73]. These instabilities may slow learning, or even destabilize training completely. Various solutions have been proposed, including freezing the embedding layer [8], adding additional layer normalization [14, 25], or reparametrizing the weights [68].

In our work we investigate instabilities which arise during CLIP training. Unlike the instabilities observed in [14, 68] which lead to a slow divergence, we study fast loss spikes. Our results indicate that these spikes arise when the second moment estimator is out of date for early layers.

While our analysis and methods build directly on Shazeer and Stern [54] (AdaFactor), there are important differences. In contrast with Shazeer and Stern [54], who only observe instabilities without warmup, we observe instabilities despite a long warmup period. Moreover, in contrast with Shazeer and Stern [54] we find that an out-of-date second moment estimator is primarily an issue for the (patch) embedding layer, and measure how well loss spikes are predicted by this event. Finally, we note that researchers have moved away from AdaFactor in its original formulation for large-scale training [47, 11, 69], finding AdaFactor to under-perform AdamW [47]. We believe this is due to the factored second moment or absence of first moment. This is why our focus is AdamW [37] which is the de facto standard optimizer for transformers.

After the initial version of this paper we became aware of Cohen et al. [12] which offers a general and principled treatment of fast loss spikes, and which we recommend to readers. Moreover, we direct the readers attention to the concurrent work of [41].

## 3.2 Experimental setup

As in Section 2, we train ViT CLIP models on LAION [53] using OpenCLIP [29] and evaluate them zero-shot on ImageNet. Since we are not interested in final performance and instead interested in studying instability—even for very large models—we use a short run which allows us to conduct multiple experiments. Concretely, we use patch-dropout 0.5 [35] and 20k iterations. The first 5k iterations are linear warmup while the remainder are cosine decay [36]. We follow the CLIP paper [46] in that i) we do not use gradient clipping unless otherwise mentioned, though we do clip the logit_scale parameter, and ii) we add a layer-norm after the patch embedding and before the main transformer. Unless otherwise mentioned, experiments use batch size 16384 (per-gpu batch size of 256), learning rate 2e-3 and weight decay 0.2. We initially tried adding a layer-norm before the patch embedding as in [34], but removed this as we found it to hurt performance at CLIP ViT-Huge scale.

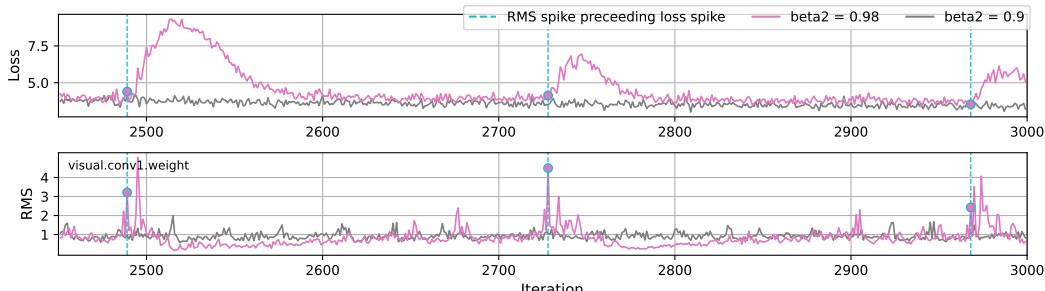

Figure 4: The learning signal can change so that the AdamW second moment estimator $u_t$ is out-of-date and underestimates the squared gradients $g_t^2$. This can be detected if the aggregate quantity $\mathsf{RMS}_t = \sqrt{\mathbb{E}\left[g_t^2/u_t\right]}$ is far from 1. This figure observes a predictive relationship between the event of an RMS spike and a loss spike—we observe a spike in $\mathsf{RMS}_t$ 1-8 iterations before a loss spike. For lower $\beta_2$, $\mathsf{RMS}_t$ does not deviate far from 1. This result looks at $\mathsf{RMS}_t$ for the patch embedding layer only. This predictive relationship is further examined in Figures 15 to 20 of Appendix E.

### 3.3    Loss spikes increase with model size, batch size, and learning rate

We begin our studying of loss spikes by observing how their presence varies when changing model size, batch size, and learning rate. The following sections build on these observations—in particular the finding that lowering the AdamW $\beta_2$ hyperparameter removes spikes entirely.

We find that loss spikes increase when increasing model size, batch size, or learning rate. The first result is shown in Figure 3 while the second two analagous results are in Appendix Figures 12 and 13. Importantly, these figures show that loss spikes can be avoided by reducing the $\beta_2$ hyperparameter for in AdamW. On the other hand, if $\beta_2$ is reduced too much then learning is slowed which results in worse performance [50].

### 3.4    On $\beta_2$ and an out-of-date second moment estimator

Based on the observation in the previous section that lowering $\beta_2$ reduces spikes, this section traces the cause of loss spikes to an out-of-date second moment estimator in the patch embedding layer.

**Overview.** Adaptive optimizers such as AdaGrad [22], Adam [33], or AdaFactor [54] scale the update differently for each individual parameter. This is often conceptualized a per-parameter learning rate. For instance, in Adam/AdamW, per-parameter updates are scaled by the inverse root of the exponential moving average of squared gradients (see the code for AdamW in Algorithm 2, ignoring for now the modifications in pink which we discuss in Section 3.5).

This adaptivity can be a very useful tool for accelerating training, but can also cause issues when the learning signal changes. Concretely, exponential moving averages can become out of date causing updates to be scaled by a value that is too large. This issue is discussed in Section 5 of Shazeer and Stern [54], and we summarize below.

As in Algorithm 2, let $u_t = \{u_{t,j}\}_{j=1}^n$ denote the exponential moving average (EMA) of squared gradients $g_t^2 = \{g_{t,j}^2\}_{j=1}^n$ for neural network parameters $\theta \in \mathbb{R}^n$. Ignoring the bias correction term[1], at each iteration $t$, $u_t$ is updated as $\beta_2 u_{t-1} + (1 - \beta_2)g_t^2$ where $\beta_2$ is referred to as the *decay* for the EMA. Then, the update is scaled by $1/\left(\sqrt{u_t} + \epsilon\right)$, where $\epsilon$ is a small value added numerical stability. Often the ratio $v_t/\left(\sqrt{u_t} + \epsilon\right)$ is thought of as signal-to-noise ratio of the gradient over time.

However, this method can break down when the learning signal changes and $u_t$ ceases to be a good estimator for the running average of $g_t^2$. Consider the case where the gradient magnitudes have been historically very small for some parameters so $1/\left(\sqrt{u_t} + \epsilon\right)$ is large for those parameters. If, then, at iteration $t$ those parameters suddenly receive a larger gradient signal the update can be catastrophically big. We refer to the scenario as the **stuck-in-the-past** scenario.

Overall, if $\beta_2$ is too small then convergence may be slowed [50]. If $\beta_2$ is too large then $u_t$ can become out-of-date and no longer a good estimator for $g_t^2$, resulting in per-parameter scaling that is too large.

---

[1]In practice, the EMA is debiased with a correction term. Algorithm 2 follows AdaFactor section 7.1 in applying the correction term to $\beta_1$, $\beta_2$. Adam is often written with the correction term applied to $v_t$, $u_t$ but they are equivalent [54].

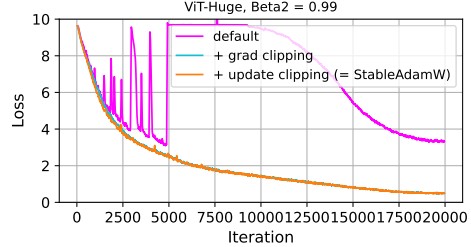 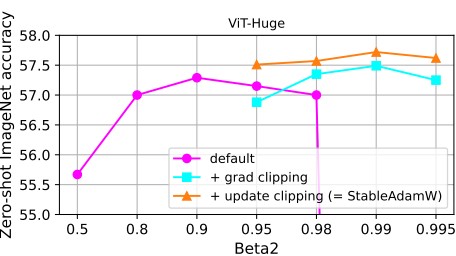

Figure 5: Adding update clipping to AdamW mitigates loss spikes and outperforms other interventions such as gradient clipping with norm 1. Code for the AdamW-AdaFactor hybrid we recommend of AdamW + update clipping is in Algorithm 2. The left plot shows loss curves for $\beta_2 = 0.99$ while the right displays accuracy ablating over $\beta_2$.

**Measurement.** We now discuss measurement of the aforementioned **stuck-in-the-past** scenario and search for a predictive relationship between this event and a loss spike. We follow Shazeer and Stern [54] and measure the following root-mean-square quantity, $\mathsf{RMS}_t = \sqrt{\mathbb{E}\left[g_t^2/u_t\right]}$. If $u_t$ is a good estimator for $g_t^2$ then the aggregate quantity $\mathsf{RMS}_t$ will be around 1. The **stuck-in-the-past** scenario described above corresponds to an $\mathsf{RMS}_t \gg 1$.

As illustrated in Figure 3 we observe instability for high $\beta_2$ in our experiments even though we have 5k iterations of warm-up. While Shazeer and Stern [54] first recognize the out-of-date second moment estimator issue, in their experimental setting they only observe instability without warm-up.

We now aim to establish a predictive relationship between the **stuck-in-the-past** scenario and loss spikes. We present initial results in Figure 4, where we examine $\mathsf{RMS}_t$ for the the visual transformer patch embedding layer, visual.conv1.weight. This means that the expectation is computed over parameters in visual.conv1.weight only. This figure illustrates a few important findings: i) loss spikes tend to follow 1-8 iterations after an RMS spike, ii) loss spikes slow learning as recovery time is required, and iii), $\mathsf{RMS}_t$ stays around 1 for lower $\beta_2$.

As this is just one example, we further elaborate on the predictive relationship between an RMS spike in the embedding layer in Section E through Figures 15, 16, 17, 18, 19, and 20. For analysis purposes, we define a heuristic to characterize loss and RMS spikes in visual.conv1.weight. We then show that 28 out of 30 detected loss spikes follow an RMS spike by 1-8 iterations, while the probability that a loss spike follows an RMS spike by chance is only 1%. Moreover, we find that the same predictive relationship does not exist for the RMS in other transformer layers.

### 3.5   StableAdamW: AdamW with update clipping from AdaFactor

This Section develops and tests StableAdamW (Algorithm 2), an AdamW-Adafactor hybrid.

To stabilize training, the AdaFactor optimizer divides the learning rate for iteration $t$ by $1/\max(\mathsf{RMS}_t, 1)$.[2] They refer to this as *update clipping*. The effect is to slow training when $u_t$ is no longer a good estimator for $g_t^2$.

As discussed in Section 3.4, our stability issues can be traced to an out-of-date $u_t$ which is what led Shazeer and Stern [54] to update clipping, even though their stability issues are also solved with warm-up. Therefore, we port update clipping to the standard AdamW optimizer with $d = 1$ and refer to the resulting AdamW-Adafactor hybrid as StableAdamW (Algorithm 2). A modification we make is to compute and divide learning rate by $\max(\mathsf{RMS}_t, 1)$ independently for each tensor, which is for implementation convenience. This means that the expectation will be computed independently for each layer to produce a different $\mathsf{RMS}_t$.

We now test how StableAdamW compares with other stability interventions such as gradient clipping[3] or lowering $\beta_2$. These results, presented in Figure 5 find that StableAdamW (i.e., AdamW + update clipping) outperforms these aforementioned interventions for CLIP ViT-Huge. While gradient clipping and update clipping both remove instability, update clipping performs better in terms of zero-shot ImageNet accuracy. With update or gradient clipping, higher $\beta_2$ such as 0.99 tends to

---

[2]They actually introduce a hyperparameter $d$ and use $1/\max(\mathsf{RMS}_t/d, 1)$, but recommend setting $d = 1$.

[3]We clip at global norm 1. We observed instability when trying 2 instead of 1. We did not tune this further, but note that 1.0 is standard in, e.g., PaLM [11], and Scaling Vision Transformers [69].

perform better. Appendix F provides further commentary and implementation considerations for StableAdamW.

# 4 Limitations, broader impacts, and conclusion

We believe the main limitation of our work is that it is non-exhaustive. For instance, we only simulate float8 training and our experiments focus solely on CLIP-style training. In terms of broader impact, our work may enable additional CLIP models, whose broader impact is examined extensively by Section 7 of Radford et al. [46]. Finally, we believe that our findings on accelerating and stabilizing large multi-modal model training will be broadly useful to the community.

**Acknowledgements**

For insightful discussions we thank Romain Beaumont, Yair Carmon, Mehdi Cherti, Brian Cheung, Alex Fang, Gabriel Ilharco, Jenia Jitsev, LAION, Sarah Pratt, Christoph Schuhmann, Ross Whightman, and Sho Yaida. We thank Emad Mostaque and stability.ai for compute resources.

This work is in part supported by NSF IIS 1652052, IIS 17303166, DARPA N66001-19-2-4031, DARPA W911NF-15-1-0543 and gifts from Allen Institute for Artificial Intelligence.

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

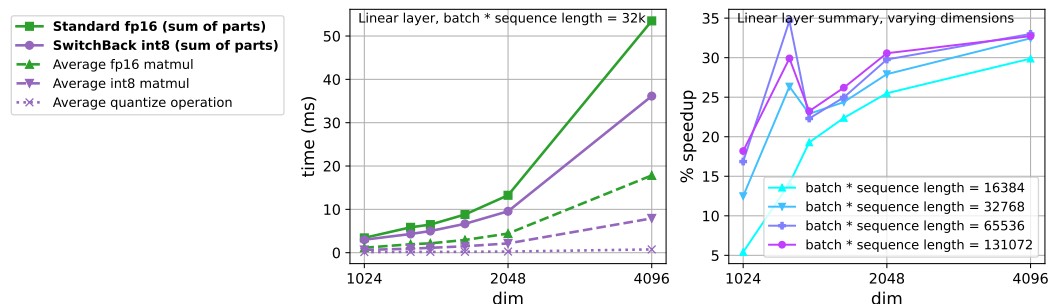

Figure 6: **(Left)** Individually profiling operations which constitute a forward and backward pass in a linear layer for i) SwitchBack using triton kernels and ii) an fp16 baseline using `torch.matmul`. Times are averaged over a linear layer from dim to $4 \cdot$ dim and a linear layer from $4 \cdot$ dim to dim—representative of the linear layers in a transformer MLP. **(Right)** The % speedup of SwitchBack over a standard fp16 linear layer when all operations in Figure 6 (left) are summed.

## A   Additional Related Work on Quantization

The literature on training neural networks in low-bit precision is vast. The main differentiating factor of our work is that we train relatively large models – in fact, we train the largest 8-bit vision transformers to date.

The literature agrees that quantization of very large networks is more difficult than for smaller networks [18, 17, 65, 24]. As such, we divide our related work into three parts: (1) large-scale low-precision neural network (larger than BERT-large), and (2) low-precision training of smaller networks.

**Large-scale Low-precision Neural Networks.**   Our work is currently the only work that does low-precision (8-bit and below) training of very large networks with more than 230M parameters. Other related work studies inference at scale. SmoothQuant [65], ZeroQuant [66], NuQmm [42], and LLM.int8() [17] study inference with Int8 matrix multiplication. Another line of work studies large models inference with more than 250M parameters by considering 16-bit inputs and k-bit weights [16, 24, 67].

**Small Scale Low-precision Training**   Training of small-scale low-precision neural networks can take many shapes and forms, such as quantization for integer only devices, quantization for mobile device, or quantization to accelerate training. One way to break up these directions is through the data type used and the neural network trained. One major direction is to quantize convolutional neural networks often for fast and memory efficient usage on edge devices [77, 6, 13, 78, 23, 76, 30]. Further work in this area is discussed in the survey by [45]. Another line of work is centered around 8-bit float data types which can be used to accelerate training of neural networks [21, 56, 62, 38, 3]. Lastly, a common application is to finetune (similar to training) BERT models to particular datasets. This not only decreases the model footprint and increases inference speed but adjusts the model to new data [2, 32, 74, 55, 75].

## B   Achieving speed-ups with SwitchBack

We now test the speedups offered by SwitchBack by first examining individual operations and then end-to-end training.

We profile all of the operations which constitute a forward and backward pass for a single linear layer in Figure 6 (left) for both SwitchBack and the baseline. For SwitchBack we profile our custom triton kernels and for the baseline we profile `torch.matmul`. Overall, we observe that int8 multiplies occupy just over half the time as standard fp16 matmuls, and that quantize operations are roughly an order of magnitude less time than a matmul. Note that our int8 matmuls are fused with the dequantize operation.

Figure 6 (right) displays the % speedup of SwitchBack over a standard fp16 layer when all operations in Figure 6 (left) are summed. Overall, the advantage of SwitchBack is greater for larger dim and batch_size $*$ sequence_length. Overall, the speedup ranges from 5% to 35%. We see a bump at dim $= 1280$ because standard PyTorch matmuls do not have optimized kernels for matrices of this

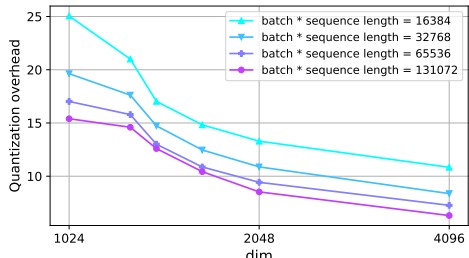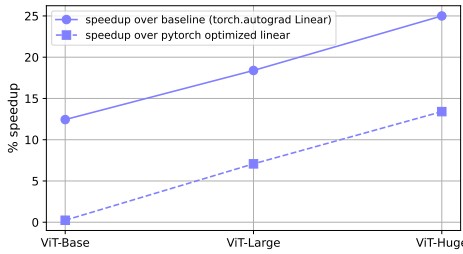

Figure 7: **(Left)** Measuring the % of time occupied by quantize operations for a SwitchBack linear layer, which is usually less than 20% and decreases with dim. **(Right)** Benchmarking speedups for end-to-end CLIP training on a single node (with 4 A100 GPUs, per-GPU batch size 256, and gradient checkpointing) for various model sizes when replacing all linear operations in the transformer with SwitchBack (i.e., key, query, value, and out projections as well as the MLP). speedups reported over i) a custom linear layer implemented with torch.autograd (Algorithm 5), which matches our implementation of SwitchBack that uses torch.autograd, and ii) using the standard PyTorch nn.Linear which includes additional background C++/CUDA optimizations which we do not replicate. LLM.int8() [17] does not provide speed-ups over the torch.autograd or nn.Linear baseline at this scale—we compare the speed of SwitchBack and LLM.int8() in Figure 8.

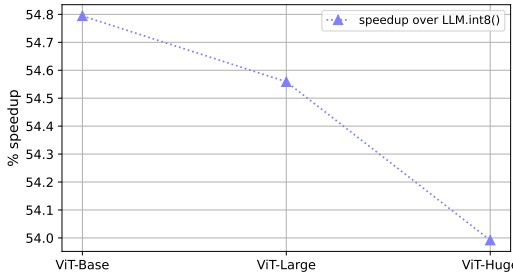

Figure 8: Benchmarking speedups of SwitchBack compared to LLM.int8() [17] for end-to-end CLIP training on a single node (with 4 A100 GPUs, per-GPU batch size 128, and gradient checkpointing) for various model sizes when replacing all linear operations in the transformer (i.e., key, query, value, and out projections as well as the MLP).

size while we use triton's autotune feature which provides fine-grained optimized kernels for matrices of any size. Our kernels are easy to modify as they are written in Triton [57], and the code to run the benchmarks and produce Figure 6 is open sourced. In doing so, we invite the community to further improve the kernels and provide a benchmark for measuring this progress. Due to computational constraints we have not tested dim > 4096 and it's possible the kernels require additional tuning to perform well at that scale.

One downside of SwitchBack is that it requires quantize operations. However, it is already evident from Figure 6 that quantize operations occupy a small amount of time compared to matmuls. This is highlighted by Figure 7 (left) which displays the fraction of time occupied by quantize operations relative to matmuls for SwitchBack linear layers. Quantize operations occupy at most 25% of the time, this fraction decreases to around 10% or below for large dim.

We now conduct end-to-end speed tests for CLIP training on a single node with 4x A100 GPUs (Figure 7, right). This is in contrast with the speedup measurements so far in this which have measured individual layers independently. We benchmark speedups relative to using i) a baseline linear layer which we implement in PyTorch with torch.autograd.linear (Algorithm 5) and ii) the PyTorch optimized linear layer nn.Linear. In both cases the speedups increase when going from CLIP ViT-Base to CLIP ViT-Huge. However, there is an additional ~12.5% speedup when comparing SwitchBack to the baseline linear layer which uses torch.autograd. We believe this comparison is fair because SwitchBack is also implemented using torch.autograd, while the standard PyTorch nn.Linear layer has additional C++ and CUDA optimizations that we do not implement. We hope to collaborate with the PyTorch team to realize the additional ~12.5% speedup. Finally, we note that the kernels from LLM.int8() [17] do not provide speedups over fp16 at the scale we consider.

# C  Additional code and figures

## C.1  Additional Code

This Section provides additional pseudocode:

- Algorithm 3 is the memory effecient variant of SwitchBack.

- Algorithm 4 is the variant of SwitchBack which uses row- and column-wise quantization for the weights. For SwitchBackQ, the forward pass is given by

$$\frac{1}{127^2}\text{state}_{\text{row}}(X)\text{state}_{\text{row}}(W)^\top * \underbrace{Q_{\text{row}}(X)\,Q_{\text{row}}(W)^\top}_{\text{int8 matmul}} \tag{4}$$

  where $*$ is an elementwise product. Again, we append `_transpose` to a function in Algorithm 4 to mean that the operation is fused with a transpose.

- Algorithm 5 is a standard linear layer implemented with torch.autograd.

---

**Algorithm 3** Memory efficient **SwitchBackM**

---

```python
class SwitchBackMMatmul(autograd.Function):
    @staticmethod
    def forward(ctx, X, W):
        # X [b, n] inputs
        # W [n, m] weights

        X_int8, state_X = row-wise_quantize(X)
        del X
        W_int8, state_W = tensor-wise_quantize(W)

        # save tensors in ctx
        ctx.save = X_int8, state_X, W_int8, state_W

        # Return output
        return matmul_int8_and_dequanitze(
            X_int8, W_int8.t(), state_X, state_W
        )

    @staticmethod
    def backward(ctx, G):
        # G [b, m] gradient to output

        # Recover tensors from ctx
        X_int8, state_X, W_int8, state_W = ctx.save

        X = dequantize_row-wise(X_int8, state_X)
        del X_int8
        W_gradient = matmul_fp16(G.t(), X)
        del X

        G_int8 = row-wise_quantize(G)
        del G
        W_int8 = W_int8.t().contiguous()

        # Use 8bit matmul only for X_gradient
        X_gradient = matmul_int8_and_dequanitze(
            G_int8, W_int8.t(), state_X, state_W
        )

        return X_gradient, W_gradient

class SwitchBackMLinear(nn.Linear):
    def forward(self, X):
        return SwitchBackMMatmul.apply(X, self.weight)
```

---

**Algorithm 4** SwitchBack with row-wise and column-wise quantization for the weights **SwitchBackQ**

```python
class SwitchBackQMatmul(autograd.Function):
    @staticmethod
    def forward(ctx, X, W):
        # X [b, n] inputs
        # W [n, m] weights

        # save tensors in ctx
        ctx.save_for_backward = X, W

        X_int8, state_X = row-wise_quantize(X)
        W_int8, state_W = row-wise_quantize(W)

        # Return output
        return matmul_int8_and_dequanitze(
            X_int8, W_int8.t(), state_X, state_W
        )

    @staticmethod
    def backward(ctx, G):
        # G [b, m] gradient to output

        # Recover tensors from ctx
        X, W = ctx.save_for_backward

        G_rowwise = rowwise_quantize(G)
        W_int8, state_W = column-wise_quantize_transpose(W)

        # Use 8bit matmul only for X_gradient
        X_gradient = matmul_int8_and_dequanitze(
            G_int8, W_int8.t(), state_X, state_W
        )
        W_gradient = matmul_fp16(G.t(), X)

        return X_gradient, W_gradient

class SwitchBackQLinear(nn.Linear):
    def forward(self, X):
        return SwitchBackQMatmul.apply(X, self.weight)
```

**Algorithm 5** A standard linear layer implemented with torch.autograd

```python
class StandardLinearMatmul(autograd.Function):
    @staticmethod
    def forward(ctx, X, W):
        # X [b, n] inputs
        # W [n, m] weights

        # save tensors in ctx
        ctx.save_for_backward = X, W

        # Return output
        return torch.matmul(X, W.t())

    @staticmethod
    def backward(ctx, G):
        # G [b, m] gradient to output

        # Recover tensors from ctx
        X, W = ctx.save_for_backward

        X_gradient = torch.matmul(G, W)
        W_gradient = torch.matmul(G.t(), X)

        return X_gradient, W_gradient

class StandardLinear(nn.Linear):
    def forward(self, X):
        return StandardLinearMatmul.apply(X, self.weight)
```

## C.2   Additional Figures

This section presents additional figures.

- Figure 9 shows the loss curves throughout training for the models in Figure 1.

- Figure 10 presents a more fine-grained version of Figure 6.

- Figure 11 shows the mean and max for the gradient and activation (i.e., feature) throughout training.

- Figures 12 and 13 respectively examine the effect of batch size and learning rate on loss spikes. They find that loss spikes increase with high batch size and learning rate, but can be mitigated by reducing AdamW $\beta_2$. However, reducing $\beta_2$ by too much can slow learning.

- Figure 14 shows that using a schedule for $\beta_2$ of the form $1 - \text{iteration}^{-\lambda}$ does not improve accuracy.

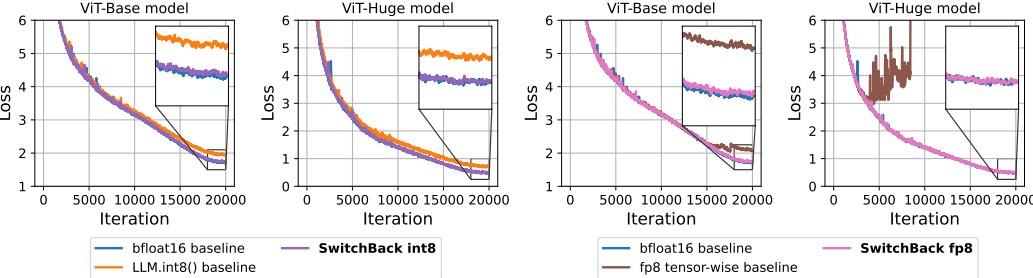

Figure 9: Loss curves for the CLIP ViT-Base and CLIP ViT-Huge models evaluated in Figure 1. The left two plots display results for int8 training while the right two plots display results for float8 (fp8) training.

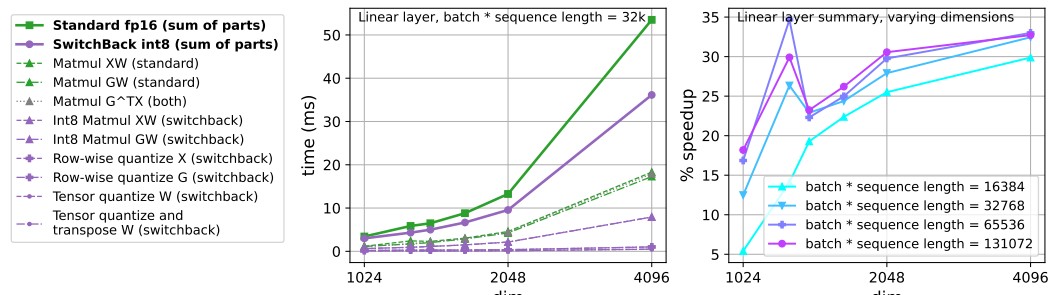

Figure 10: A more fine-grained version of Figure 6.

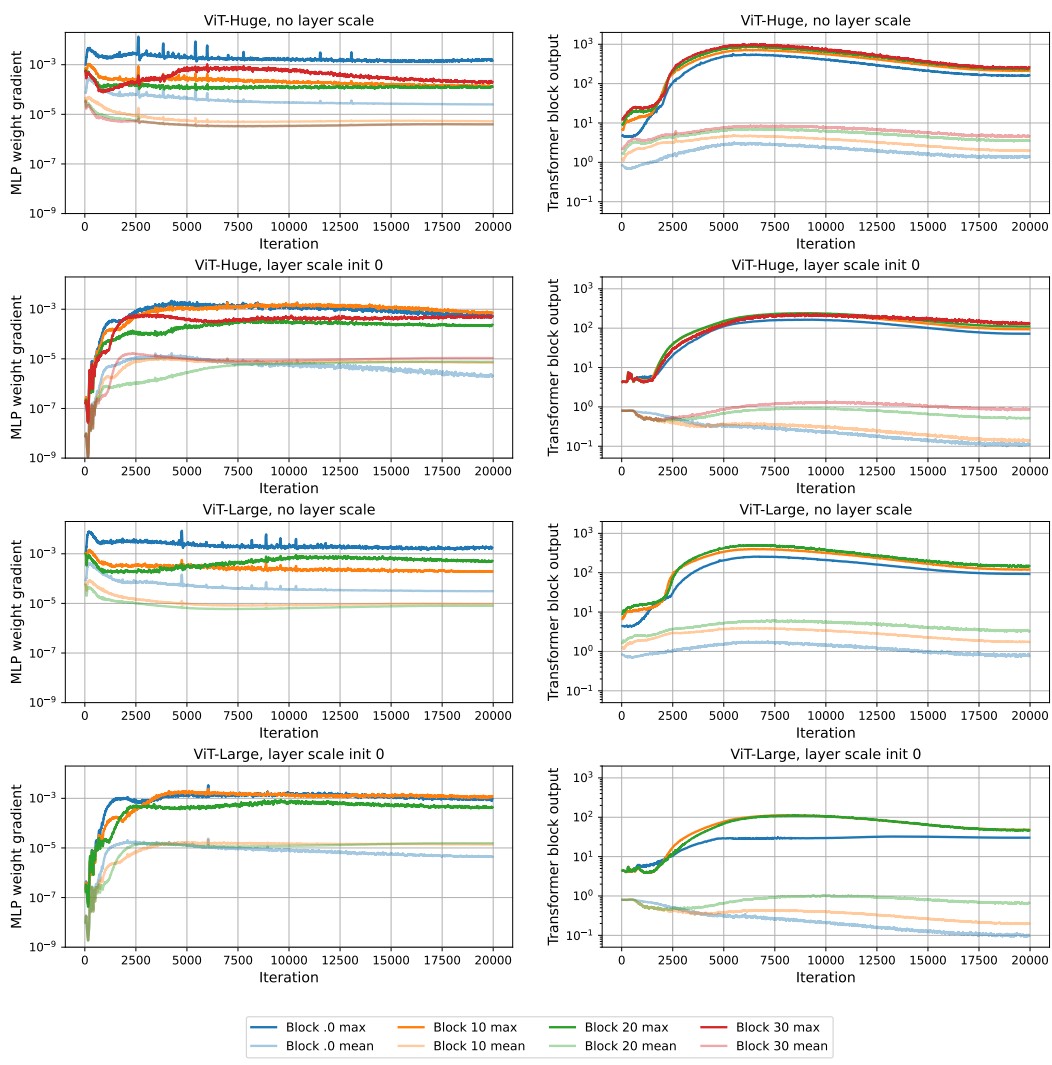

Figure 11: The mean and max for (left) the gradient to the MLP weight and (right) the output of a transformer block throughout training. Different rows correspond to different choice of model size and layer scale.

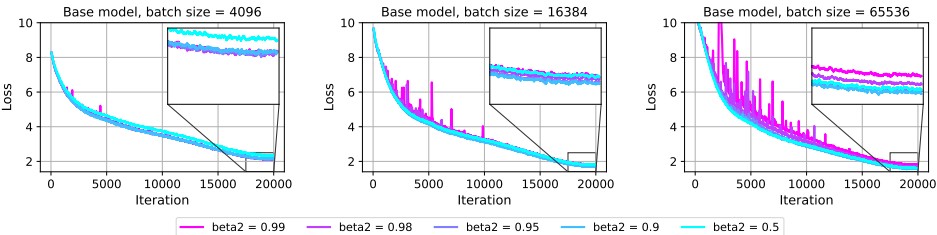

Figure 12: Loss spikes increase with **batch size** for fixed learning rate and model size. Reducing AdamW $\beta_2$ from its default in PyTorch of 0.999 mitigates loss spikes. Reducing $\beta_2$ too much slows training.

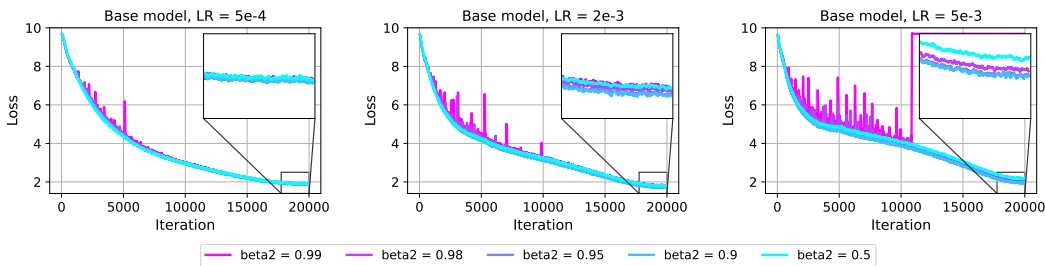

Figure 13: Loss spikes increase with **learning rate** for fixed batch size and model size. Reducing AdamW $\beta_2$ from its default in PyTorch of 0.999 mitigates loss spikes. Reducing $\beta_2$ too much slows training.

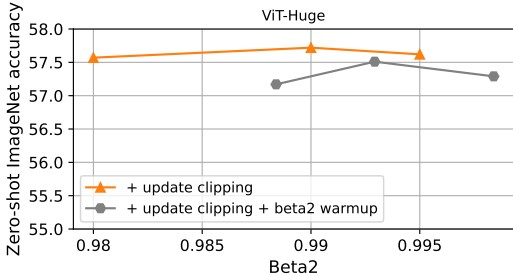

Figure 14: We try a schedule for $\beta_2$ which is used in AdaFactor [54] and PaLM [11] and refer to the experiment as $\beta_2$ warmup. This means that $\beta_2$ at iteration $k$ is $1 - \text{iteration}^{-\lambda}$. In this Figure we try $\lambda = 0.45, 0.5, 0.65$ and show on the $x$-axis $\beta_2$ at the final iteration. This $\beta_2$ warm-up does not improve accuracy in our setting.

# D   Analysis

Consider a matrix multiplication $UV$ for $U \in \mathbb{R}^{n \times k}$ and $V \in \mathbb{R}^{k \times m}$. This matmul consists of computing inner products between vectors of length $k$.

This section shows that error due to quantization increases with $k$. This suggests why SwitchBack may achieve high accuracy, as we avoid quantizing matmuls for which $k$ is very large. For the weight gradient computation, which we leave in high precision, $k$ is batch size times sequence length, which is often $\approx 32000$ in our experiments. For the other operations which comprise a matmul, $k$ is less than $4 \cdot \text{embed\_dim}$ which is $\leq 8000$ in our experiments. These dimensions are standard for CLIP training experiments [46, 9].

## D.1   Analyzing variance due to quantization for inner products

This section measures the variance due to quantization for the inner product between $u$ and $v$. Let $u$, $v$ be vectors of length $k$ vectors with each element drawn i.i.d. from a distribution with mean 0. Let $u_i$ have variance $\sigma_u^2$ and $v_i$ have variance $\sigma_v^2$.

Next, let $\hat{u}$ and $\hat{v}$ be the quantized versions of $u$ and $v$, respectively. We model quantization error as $\hat{u}_i = u_i + \epsilon_i$ and $\hat{v}_i = v_i + \xi_i$ where $\epsilon_i, \xi_i$ are i.i.d. mean centered random variables with variance $\sigma_q^2$.

The aim of this section is to show that variance due to quantization grows with $k$. Our analysis is conservative because we do not assume the variance of $\epsilon_i, \xi_i$ increase with $k$, though in practice we believe they would as the absmax of $u$ and $v$ increases with $k$.

We first examine the variance of $\hat{u}_i \hat{v}_i$. By using that all random variable are mean centered, this variance is given by,

$$\text{Var}(\hat{u}_i \hat{v}_i) = \mathbb{E}\left[(\hat{u}_i \hat{v}_i)^2\right] \tag{5}$$

$$= \mathbb{E}\left[((u_i + \epsilon_i) \cdot (v_i + \xi_i))^2\right] \tag{6}$$

$$= \mathbb{E}\left[(u_i v_i + \epsilon_i v_i + \xi_i u_i + \epsilon_i \xi_i)^2\right] \tag{7}$$

$$= \mathbb{E}\left[u_i^2 v_i^2 + \epsilon_i^2 v_i^2 + \xi_i^2 u_i^2 + \epsilon_i^2 \xi_i^2\right] \tag{8}$$

$$= \text{Var}(u_i v_i) + \sigma_q^2(\sigma_u^2 + \sigma_v^2 + \sigma_q^2). \tag{9}$$

Next, we use linearity of variance for independent random variables to calculate $\text{Var}\left(\langle \hat{u}, \hat{v} \rangle\right)$. This is given by,

$$\text{Var}\left(\langle \hat{u}, \hat{v} \rangle\right) = \sum_{i=1}^{k} \text{Var}(\hat{u}_i \hat{v}_i) \tag{10}$$

$$= \sum_{i=1}^{k} \text{Var}(u_i v_i) + \sum_{i=1}^{k} \sigma_q^2(\sigma_u^2 + \sigma_v^2 + \sigma_q^2) \tag{11}$$

$$= \text{Var}\left(\langle u, v \rangle\right) + k \cdot \sigma_q^2(\sigma_u^2 + \sigma_v^2 + \sigma_q^2). \tag{12}$$

## D.2   Takeaways

We have shown that for inner products with length $k$ vectors, variance due to quantization increases with $k$. This means the variance of output units/features due to quantization increases with $k$ which can thought of as making the outputs more noisy. Noise compounds throughout the network and will eventually drown out useful signal—for large $k$ the network features or gradient will no longer lead to effective learning.

## D.3   Why LLM.int8() fails: LLMs vs CLIP models

This Section details our hypothesis for why SwitchBack outperforms LLM.int8() for CLIP training, which is conditioned on the analysis in Section D.1 being a good model for training.

From our analysis we have shown that the variance in the output features increases with the size of the inner products of a quantized matrix multiplication compared to the full precision matrix multiplication. As such, we may have different failure modes for transformers pretrained on text, such as GPT-3 [5] or LLaMA [59], compared to CLIP models [46].

Pretrained large language models (LLMs) tend to have larger weight matrices relative to their batch sizes when compared to CLIP models. CLIP models perform best when the batch size is large [46, 44, 9]. As a consequence, LLMs and CLIP models have their most noisy operations for different matrix multiplications. LLMs are most noisy in the forward pass $XW^T$ and during layer-to-layer back propagation $\dot{Y}_k W_k = \dot{X}_{k-1}$ where inner product dimension are large, for example, they are 32768 and 8192 for the output projection of LLaMA 65B, 32768 and 8192. While the weight gradient inner product size is determined by the per-GPU batch size, which is 2048 for LLaMA [59] (4M tokens per full batch distributed across 2048 GPUs). As such, if the quantization produces the same variance in quantization errors, then the weight gradient in LLM int8 training is between 4x and 16x less noisy if the analysis in Section D.1 is a good model for training.

For CLIP training with ViT-Huge, we have a batch size of 65536 per GPU (256x images of size 224x224 inputs with patch size 14x14, leading to 16x16 patches for each images, resulting in 65536 patches per GPU). The dimensions for the weight matrices are $1280 \times 5120$. As such, analogous to above for the LLaMA LLM, the weight gradient in CLIP models is between 51.2x to 12.8x more noisy compared to the forward and layer-to-layer backpropagation operations if the analysis in Section D.1 is a good model for training. Notice that the CLIP weight gradient is twice as noisy compared to the most noisy LLaMA 65B operations if we assume that all quantization operations have the same error variance.

As such, low-precision LLM training and CLIP requires high-precision quantization routines for different parts of the training.

This also gives the reason why we believe LLM.int8() fails despite replicating inference performance – the weight gradient in CLIP training is a highly noisy operation which might not give enough signal to SGD to converge to a local minimum.

# E    RMS Spikes precede Loss Spikes

This section further elaborate on the predictive relationship between an RMS spike in the embedding layer and a loss spike as in Figure 4.

We define a heuristic to characterize loss and RMS spikes which we use for analysis. We determined these heuristics by checking if they qualitatively coincided with what appeared to be a loss spike. We display results in this Section so that the reader can also evaluate if these heuristics appear reasonable.

We define RMS spikes events as $\{t : \text{RMS}_t \geq 2.3\}$ while loss spike events are defined as the set of $t$ where loss at time $t$ exceeds the running mean by 3.2 times the running standard deviation. Finally, we ignore the first 1000 iterations when learning rate is low.

We also deduplicate the RMS and loss spikes iterations as follows: multiple spikes over a short time interval of 10 iterations are only counted as one spike and start at the earliest time. Moreover, we only count a loss spike if there are multiple deviations in an interval of 10, which indicates that loss has meaningfully spiked.

Our results are as follows:

- Figure 15 observes that out of 15 total loss spikes for ViT-Huge across different $\beta_2$, 14 out of 15 come 1-8 iterations after an RMS spike in the patch embedding layer (module.conv1.weight). With only 76 total RMS spike events, the probability that a loss spike follows 1-8 iterations after an RMS spike by chance is $< 1\%$.

- Figure 16 repeats this analysis for ViT-Large, wherein 13 out of 15 loss spikes follow an RMS spike by 1-8 iterations. The probability that a loss spike follows an RMS spike by chance is $1.0\%$.

- Figure 17 zooms in on Figure 15 to show additional detail.

- Figures 18 and 19 examine the cases where loss spikes fail to be detected in Figures 15 and 16, finding them to mainly be issues with the heuristic identifying loss spikes, i.e., false positive loss spikes.

- Finally, Figure 20 repeats Figure 15 but examines the RMS of a random layer in the middle of the transformer—not the patch embedding layer. In this case, *none* of the loss spikes follow RMS spikes.

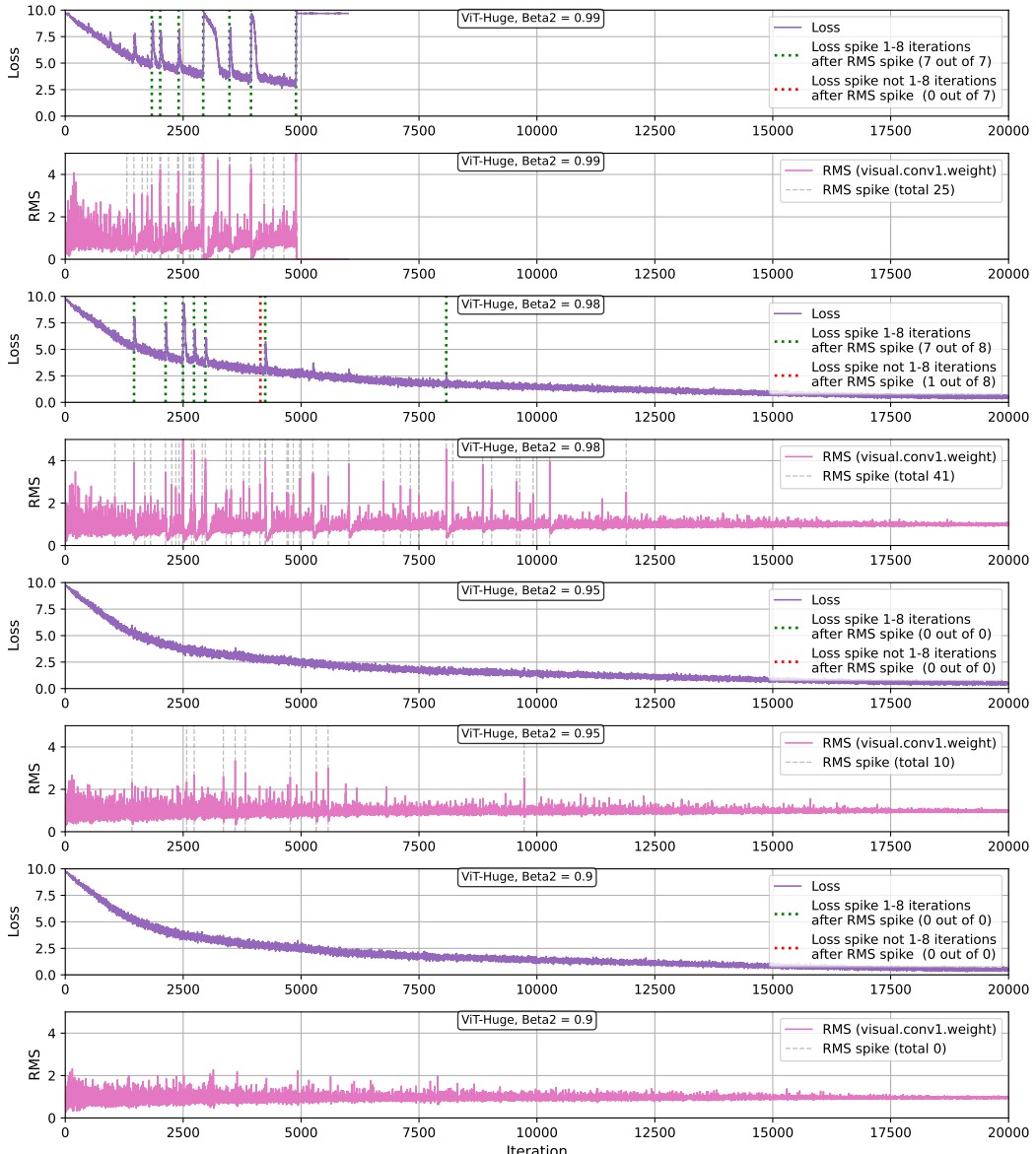

Figure 15: Observing a predictive relation between RMS spikes and loss spikes. For CLIP ViT-Huge and multiple $\beta_2$ values, we use heuristics (Appendix E) to automatically identify loss spikes which we use for analysis. Out of 15 total loss spikes, 14 follow an RMS spike in the patch embedding layer (RMS > 2.3) by 1-8 iterations. We show loss spikes which are identified by our heuristic. We use green if they follow an RMS spike and otherwise use red. An RMS Spike indicates that the second moment estimator is out of date (see Section 3.4 and Shazeer and Stern [54]). The chance that a loss spike follows 1-8 iterations after an RMS spike by chance is < 1%.

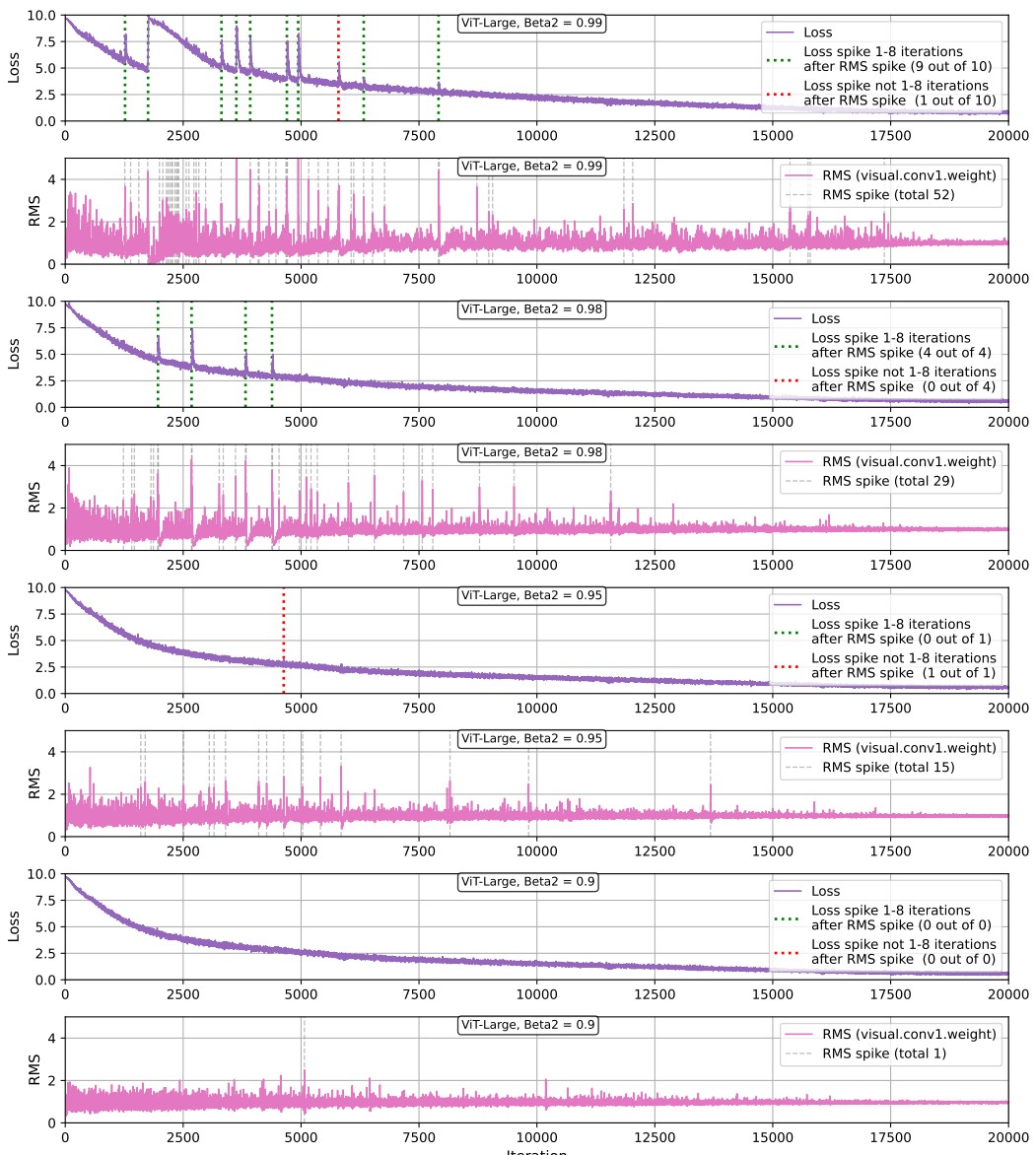

Figure 16: Observing a predictive relation between RMS spikes and loss spikes. For CLIP ViT-Large and multiple $\beta_2$ values, we use heuristics (Appendix E) to automatically identify loss spikes which we use for analysis. Out of 15 total loss spikes, 13 follow an RMS spike in the patch embedding layer (RMS > 2.3) by 1-8 iterations. We show loss spikes which are identified by our heuristic. We use green if they follow an RMS spike and otherwise use red. An RMS Spike indicates that the second moment estimator is out of date (see Section 3.4 and Shazeer and Stern [54]). The chance that a loss spike follows 1-8 iterations after an RMS spike by chance is 1.0%.

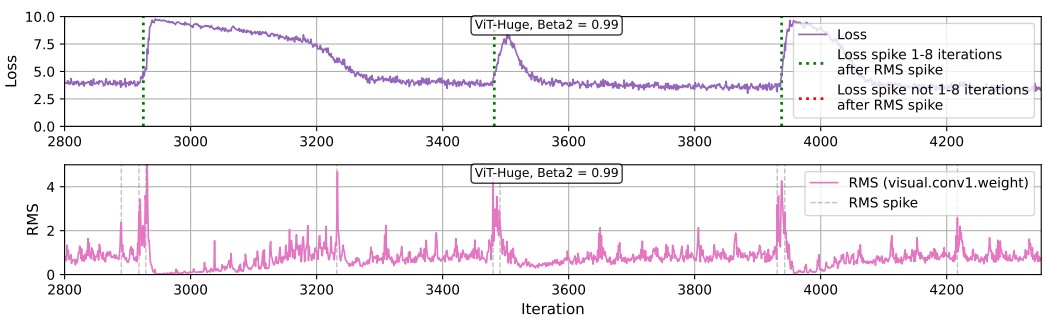

Figure 17: Zooming in on a section of Figure 15.

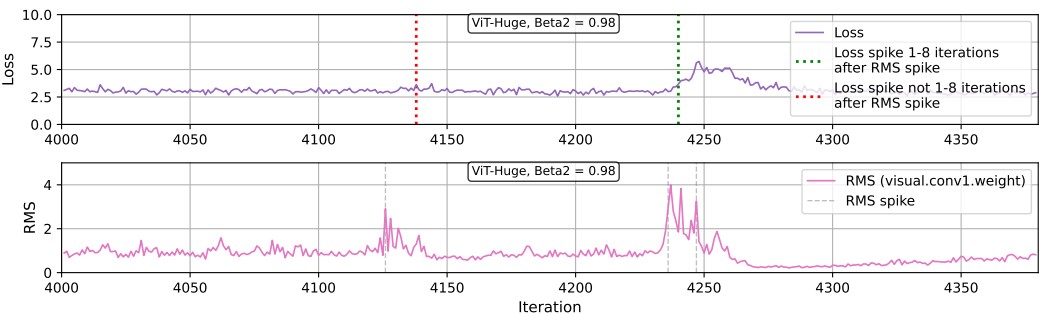

Figure 18: Examining the "failure" case in Figure 15. We believe this is not really a failure as the non-predicted red loss spike does not really appear to be a spike at all. However, adjusting our heuristic led to the issue of true spikes not being identified.

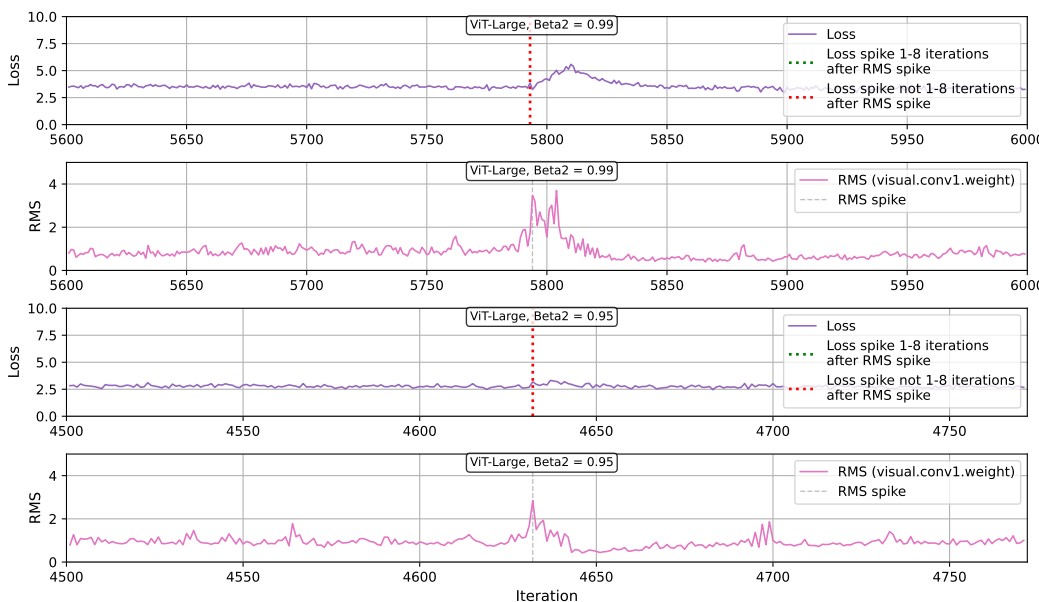

Figure 19: Examining the "failure" cases in Figure 16. We believe these to primarily issues with our heuristic, but adjusting our heuristic led to other issues such as true spikes not being identified.

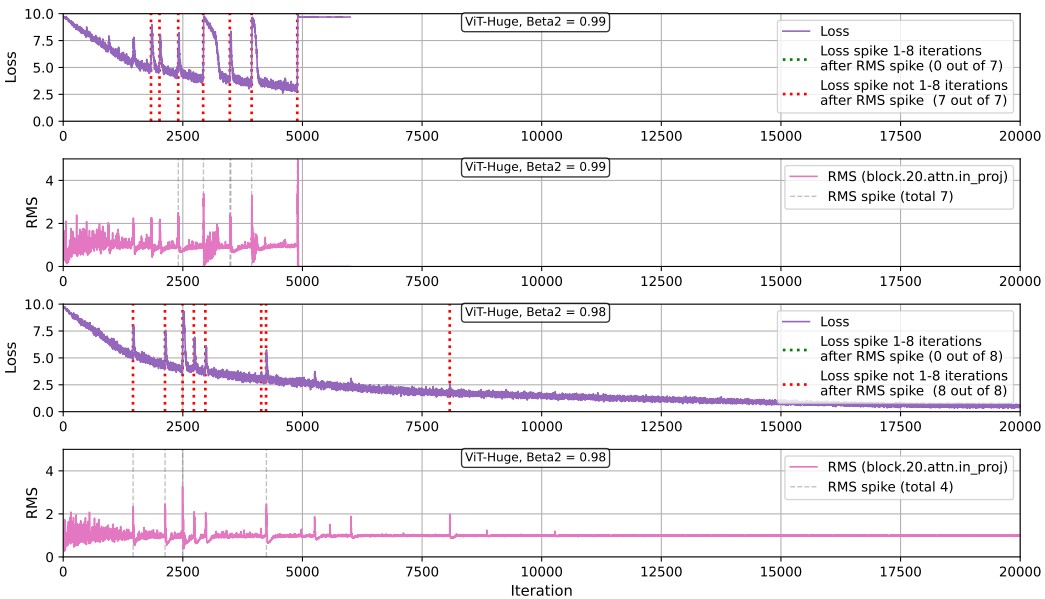

Figure 20: This figure repeats part of Figure 15 but examines the RMS of a random layer in the middle of the transformer—blocks.20.attn.in_proj—not the patch embedding layer. RMS spikes no longer precede loss spikes.

## F   StableAdamW continued

### F.1   Q&A

This Section asks and answers a series of questions the reader may have concerning Section 3.5.

- First, why not just use AdaFactor? The answer is that the community has moved away from AdaFactor [11] as they find that AdaFactor under-performs AdamW at scale [47]. We believe this is likely due to the factored moments, and not other features such as update-clipping. The goal of this work is to advocate using a hybrid. We tried porting other features from AdaFactor to AdamW such as the $\beta_2$ schedule but did not find them to help (Figure 14). Moreover, while PaLM uses an AdaFactor-AdamW hybrid, we believe they don't use update clipping.

- Another question is, why not use an optimizer such as Lion [7] which does not divide updates by any value, and is therefore immune to the stuck-in-the-past scenario. We believe this may be a promising path forward. However, while we observe that Lion outperforms AdamW at small scale, Lion still slightly under-performs AdamW for CLIP ViT-Huge scale in our experiments.

- A final question is, why consider $g_t^2$ in the numerator for computing $\mathsf{RMS}_t$ and not $v_t^2$? We also tried $v_t^2$ and found the performance worse.

### F.2   Implementation considerations

To prevent divide by 0 issues when computing $\mathsf{RMS}_t$ we compute $\mathsf{RMS}_t = \sqrt{\mathbb{E}\left[g_t^2/\mathsf{maximum}(u_t, \epsilon^2)\right]}$ where $\epsilon$ is the AdamW hyperparamer for which we use 1e-6 and maximum is an elementwise maximum. This is instead of $\mathsf{RMS}_t = \sqrt{\mathbb{E}\left[g_t^2/u_t\right]}$.

## G   Loss spikes and the loss scalar

This Section ties the low precision training results 2 with our investigation into stability. Overall we find that loss spikes can co-occur with large activations and gradients. Large activations and gradients may cause issues during low precision training due to a more limited representible range. Therefore, reducing loss spikes is an important step for successful low precision training.

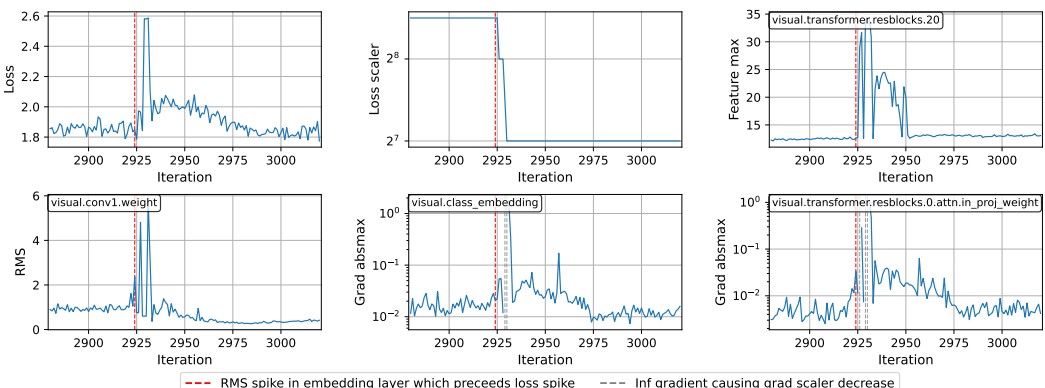

Figure 21: Avoiding loss spikes is helpful for low precision training. As shown in this figure, loss spikes can coincide with with activation spikes and gradient spikes. Large activations/gradients can cause issues during low precision training due to a more limited representible range [17].

Supporting data is illustrated by Figure 21, in which an RMS spike precedes a loss spikes which coincides with spikes in the activations (i.e., features) and gradients. As we've previously seen (Figure 2), high feature magnitudes can pose challenges for low-precision training. Moreover, the spikes in the gradient are so large that Inf/NaN values occur, which results in the loss scalar [40] dropping many times. There are a few takeaways from this observation.First, reducing loss spikes is an important step to enabling low-precision training. Second, spikes in gradient magnitude can be transient and therefore we may be adjusting the loss scalar too often—if using the PyTorch default loss scalar, thousands of iterations would be required before the loss scalar recovered to its value before this event. Finally, the layers highlighted in this figure are the main layers where Inf/NaN are encountered. Concretely, while we only track every tenth block, we never observe any Inf/NaN for any transformer block greater than 0. However, with the PyTorch default loss scalar an Inf/NaN in a single layer will skip the update for the whole network.

This motivates the loss scalar that we use in our experiments when one is required (except for in Figure 21). We use a loss scalar which i) checks for Inf/NaN at the individual tensor level and skips the update at the tensor level—not globally, and ii) remains fixed at its initial value.

This scalar allows fp16 mixed precision training for CLIP models at ViT-Huge scale where previously the scalar became too low and training diverged [9]. We also believe an adaptive block-wise scalar as in Ramesh et al. [48] would remedy this issue. One interesting remark is that often when we observe an Inf/NaN, it is in the patch embedding layer. Therefore, in the case where Inf/NaN's happen frequently it recovers the stability solution of Chen et al. [8] which is to freeze the embedding layer. As a final remark, we note that loss spikes do not always cause the loss scalar to drop, and emphasize the loss scalar can drop for various other reasons than spikes. Figure 21 is just an existence example that loss spikes can result in activation spikes and Inf/NaN gradients.

