# OpenReview forum: "Stable and low-precision training for large-scale vision-language models"
_NeurIPS.cc/2023/Conference — NeurIPS 2023 poster_

### Official Review · Reviewer_35W5 · 2023-07-01

**Soundness:** 4 excellent
**Presentation:** 4 excellent
**Contribution:** 4 excellent
**Rating:** 8
**Confidence:** 2

**Summary:**

The paper proposes methods to improve the training of large, vision-language models. On the one had, the authors propose a new linear layer for int8 quantized trainings, dubbed SwitchBack. On the other hand, a new optimizer is presented, StableAdamW, which results from combining AdamW with the update clipping technique proposed by AdaFactor [52]. These two simple but effective measures allow for faster and more stable training of large vision-language models.

**Strengths:**

- Simply having to use 8 bit precision for the first two matrix multiplies of a linear layer (forward pass to computer output + backward pass to compute gradients of the input) is a powerful way to achieve notable speed-ups while keeping the same performance as when using 16-bit training. I find this a very insightful contribution that can accelerate training for many projects that leverage transformer architectures.
- The contributed StableAdamW (AdamW + update clipping from AdaFactor) addresses instabilities that can arise during training of large vision-language models. This is a well-known issues in the community, which could have major impact for future research. The fact that StableAdamW outperforms other stability control measures such as gradient clipping or setting a lower β2 already opens up the door for very easy stability fixes across the board for vision-language models, at least for those CLIP-based.
- Detailed comparisons and results are presented.

**Weaknesses:**

- Could be interesting if the authors briefly discussed whether the results found in this paper generalize to downstream tasks, e.g., object detection or semantic segmentation. In other words, can other researchers directly apply SwitchBack layers in other transformers architectures, such us OneFormer? Can we also directly use StableAdamW? This is mentioned in the limitations section, which is appreciated. But any further points on this could be valuable to the community, as many of the breakthroughs in classification tasks have been found not to always easily translate into gains on downstream tasks.

**Questions:**

- Minor typo in line 282? “for in AdamW.”
- I assume that code will be released. Could the authors confirmed? If the results presented here are indeed generalizable to vision-language models in general, it could have major implications for the advancement of the community.

**Limitations:**

Limitations have been addressed.

---

> ### Author Rebuttal · Authors · 2023-08-07
>
> Thank you very much for your thoughtful comments and thorough review.
>
> - Weakness 1: We agree that the paper would benefit from a discussion on whether we can expect the results to generalize to downstream applications. Our analysis in Section D of the supplementary material suggests that the approach is best applied when the inner dimension of the matrix multiplication for the weight gradient computation is large, which is typically the case for large batch training. Moreover, we believe that one of the benefits of CLIP style models is that they can be used in downstream applications without modifying the weights. That is, we believe that our models may be used for a drop-in replacement in stable diffusion to provide speed-ups. We will revise the paper to elaborate on this discussion.
> - Question 1: Thank you very much for catching this, it is indeed a typo.
> - Question 2: We confirm that code will be released. Indeed, we have focused on making our triton kernels clear and hackable. While we have optimized our kernels to the best of our ability, we believe that by open-sourcing them and making them hackable, the community will be able to iterate and further the already observed speed-ups.

---

> > ### Comment · Reviewer_35W5 · 2023-08-12
> >
> > Thank you. It sounds good. Appreciate the effort to make this available to the community.
> >
> > Please fix the typos and add the proposed clarifications.
> >
> > Thanks!

---

> > > ### Author Response · Authors · 2023-08-14
> > >
> > > Of course, thank you again for your helpful comments and thorough review.

---

### Official Review · Reviewer_Nme8 · 2023-07-05

**Soundness:** 3 good
**Presentation:** 3 good
**Contribution:** 2 fair
**Rating:** 6
**Confidence:** 3

**Summary:**

The paper is a study of different ingredients necessary to train an int8-quantized CLIP model. They mainly address two aspects:
  1. quantization: they built on top of techniques previously applied to LLM inference (LLM.uint8) and expand it to the CLIP setting to optimize both inference and training; the 13-25% speedup of training and negligible (~0.1%) quality drop is achieved by carefully selecting which parts of the model are quantized to 8 bits (forward pass and part of backward pass) and which to 16 bits (part of backward pass),
  2. stabilization: they observed spikes in the loss and correlate it with times where the state of Adam optimizer gets outdated; this is mitigated by applying the same trick to Adam which was already established in the literature to stabilize AdaFactor


**Strengths:**

1.  The authors manage to train the biggest quantized CLIP up-to-date, with negligible quality drop. Learning that this is possible to do is a valuable contribution.
2.  The source of training instability at scale is identified and characterized, and a way to mitigate it is proposed.


**Weaknesses:**

The proposed quantization method targets linear layers, which are “usually >90% of compute in standard transformer models”. However, it seems that there is significant complexity introduced for a relatively small speedup of 13-25%. Was that expected? What are the remaining bottlenecks which remain to be solved? It would be interesting to get more details around this topic.

**Questions:**

1.  Is the training instability present only when quantization is used, or even without it?
2.  How to explain that loss curves for grad clipping and adam clipping are the same, but there is a difference in imagenet zero-shot accuracy in Table 5? Is this difference repeatable over several runs, or could it be measuring noise?


**Limitations:**

The authors have adequately addressed the limitations.

---

> ### Author Rebuttal · Authors · 2023-08-07
>
> Thank you very much for your thoughtful review. We hope the following addresses your interesting questions and comments.
>
> - Weakness 1: We believe there are a number of reasons that the speed-up is only 13-25%. For one, 16-bit baselines have had years of hardware support and optimization, while the same is not true for eight bit formats. We believe that more speedups can be expected as hardware support improves. For example, initial data from H100 GPUs indicates that 8-bit speedups are much higher for these GPUs, but we did not include these benchmarks since we have no access to H100 GPUs. In addition, as we illustrate in Figure 6 and 10 of the supplementary material, the quantization operations (i.e., changing from 16 bit to 8 bit precision) incur overhead which reduces the overall speed-up. However, as we show in Figure 7 of the supplementary material, this overhead decreases in proportion at scale.
> - Question 1: The instabilities we discuss in Section 3 are present even without quantization.
> - Question 2: We believe that this is not noise because we consistently observe a difference across all four values of beta2 that we try as illustrated in Figure 5 (right).

---

> > ### Comment · Reviewer_Nme8 · 2023-08-14
> >
> > Thank you for the clarifications.

---

### Official Review · Reviewer_rd6a · 2023-07-06

**Soundness:** 3 good
**Presentation:** 3 good
**Contribution:** 2 fair
**Rating:** 6
**Confidence:** 3

**Summary:**

This paper introduces an efficient and stable INT8 training method for models similar to CLIP. The proposed method offers a training latency improvement of 10-20%, which could account for a significant portion of training costs for larger models. The authors leverage LLM.int8() kernels for training, taking into account the quantization challenge for weights. Additionally, the paper explores the potential issues associated with FP8 training for such models.

**Strengths:**

- The paper is beneficial to the community due to the rising interest in low-precision training, such as INT8 and FP8. In particular, understanding how FP8, which is supported by H100 GPUs, performs across different models is essential. This work provides useful insights into these topics.
- The authors propose system/hardware-aware methods, including open-sourcing triton implementations and fused kernels, which could prove valuable for many researchers.
- The zero-init layer-scale method, a simple yet effective approach to tensor-wise quantization, is proposed.
- The systematic study of loss spike offers many points for consideration and deeper analysis.
- The introduction of the novel predictive power of patch embedding RMS is noteworthy.

**Weaknesses:**

- The paper's SwitchBack approach seems to lack novelty, as it simply observes the correlation of inner dimensions with quantization noise. The authors could explore more ingenious solutions to tackle the large inner dimension of weight gradient matmul, rather than just resorting to higher precision computing.
- Figure 1 raises a concern as LLM.int8() might not serve as an adequate baseline. LLM.int8 was designed for LLMs, not models like CLIP. Additional PTQ baselines, including MRE methods, should be considered for int8 training.
- There's a fundamental curiosity about the applicability of low-precision training like INT8/FP8. While we can establish an FP16 baseline for well-known models, it becomes challenging when we train newly updated models or datasets with no baseline. If the training fails, it's unclear if the failure stems from precision issues or model issues. Although this paper's analysis and proposal could assist in these processes, the argument still stands: why use low-precision training if stable results can be obtained with slightly slower speeds? Because this open question applies to all low-precision training and not solely to this paper, I will respect feedbacks from other reviewers in this aspect.

**Questions:**

included in weaknesses

**Limitations:**

included in weaknesses

---

> ### Author Rebuttal · Authors · 2023-08-07
>
> Thank you very much for your insightful comments. We hope the following addresses any concerns.
>
> - Weakness 1: We absolutely agree that resorting back to higher precision is a weakness and a more clever technique could alleviate this. We thank you for highlighting this as we believe it’s one of the most promising avenues for future research, and will revise the paper to indicate this shortcoming.
> - Weakness 2: Our focus is on faster training via quantization and not quantization after training. Appendix E of the LLM.Int8() paper establishes it as a practical method for large scale quantized training, though it is commonly regarded as a PTQ technique.
> - Weakness 3: Thank you very much for this very insightful question about why quantized training is even warranted. Indeed, we believe that this is very much an open question. However, if the current scaling trend persists for more years, it is likely that training runs in the future will cost upwards of 100 million US dollars. Therefore, any reduction in training time could be the difference between whether a run is feasible or not. Given the potential savings, we believe institutions will (and perhaps already are) taking the risk.

---

### Official Review · Reviewer_sdBF · 2023-07-07

**Soundness:** 3 good
**Presentation:** 3 good
**Contribution:** 3 good
**Rating:** 6
**Confidence:** 2

**Summary:**

This paper proposed methods for accelerating and stabilizing CLIP training. To accelerate training, the authors proposed the SwitchBack method, which quantizes the precision to int8 for the first two matrix multiplies but switches back to higher precision for the weight gradient. The proposed method speeds up 13-25% CLIP ViT-Huge training. To stabilize training, the authors proposed to combine the AdamW and the Adafactor optimizers and call the proposed new optimizer StableAdamW. This is observed from the experiments that when the squared gradients become underestimated by AdamW's second-moment estimator, loss spikes will occur after several iterations.

**Strengths:**

The motivation for the ideas is very clear. To train large-scale language-vision models, speed is very critical. Training stability is also very important because we need to train with large-scale data without performance degradation. These two factors are essential for training large-scale language-vision models.

The authors proposed the SwitchBack method that quantizes the precision to int8 for fast matrix multiplication and then transforms back to the original floating point precision for the weight gradient. The authors provide PyTorch torch illustrating this process in Algorithm 1. Experiments show that the proposed method could speed up 13-25% of CLIP ViT-Huge training.

The authors proposed the StableAdamW optimizer for training large-scale language-vision models. The authors observed that if the squared gradients are underestimated by AdamW's second-moment estimator, then the loss spike would occur in the subsequent few iterations. The authors proposed to combine the AdamW and the Adafactor to get their proposed StableAdamW optimizer. Experimental results show that StableAdamW stabilizes training and helps the model achieve higher zero-shot performance than the AdamW optimizer and the AdamW optimizer with gradient clipping.

**Weaknesses:**

The experiments are only conducted for the CLIP training. It's unknown whether the proposed SwitchBack and StableAdamW will work for other language-vision pretraining models such as BEiT v2, BEiT v3, or BLIP. Also, since CLIP only uses contrastive loss, it's not clear whether the proposed methods will work for other losses.

It's unclear whether the authors used the global contrastive loss computed across all GPUs or the local contrastive loss computed on each GPU. The global contrastive loss also affects speed, accuracy, and stability.

**Questions:**

$x_n$ should be $x_b$ in Eq. 1?

Why only examine RMSt for the visual transformer patch embedding layer, visual.conv1.weight?

Why not compare SwitchBack with the quantization method in [1]?

Reference:

[1] Bai et al., Towards Efficient Post-training Quantization of Pre-trained Language Models, NeurIPS 2022.

**Limitations:**

Mentioned in Weakness.

---

> ### Author Rebuttal · Authors · 2023-08-07
>
> Thank you very much for your insightful review and careful attention to details. We hope the following answers your questions.
>
> - Question1: Regarding the typo in equation 1. Thanks very much for this catch, this is indeed a typo.
> - Question 2: We find that the patch embedding layer is the source of instability, which also aligns with the findings of [1]. When we create the same plot for other layers, we do not observe the predictive relationship which we observe with the patch embedding. More information on this point is provided in L741 of the supplementary material, in particular Figure 20 of the supplementary material examines RMS and loss for a non-embedding layer.
> - Question 3: Thank you for highlighting [2], we will revise to include this important reference. However, [2] focuses on post-training quantization, i.e., quantizing after training, while our paper focuses on training in low precision. Training in low precision is a much more difficult problem.
> - Weakness 1: We agree that focusing on CLIP is a limitation, but also believe that large language-vision models are increasingly important as they underlie generative models and zero-shot methods, and that these models present a unique set of challenges. We will revise the paper to make this shortcoming more apparent.
> - Weakness 2: We use the global contrastive loss, and will revise to include this information.
>
> [1] https://arxiv.org/abs/2104.02057

---

> > ### Comment · Reviewer_sdBF · 2023-08-14
> > **Response to rebuttal**
> >
> > Thank the authors for the reply! Most of my questions are addressed.

---

> > > ### Author Response · Authors · 2023-08-14
> > >
> > > Thank you again for the constructive comments. Please don't hesitate to ask further questions if they come up. Also, if you feel it is warranted after our response we would of course appreciate if you consider raising your score.

---

### Official Review · Reviewer_AbFj · 2023-07-07

**Soundness:** 3 good
**Presentation:** 2 fair
**Contribution:** 1 poor
**Rating:** 3
**Confidence:** 5

**Summary:**

The authors introduce new methods for  accelerating and stabilizing training for large language-vision models. For acceleration, SwitchBack is proposed, which use high-precision for backwardd pass to compute the gradients for the weights. For stability, the introduce an
AdamW-Adafactor hybrid (StableAdamW).

**Strengths:**

1. This work aims to accelerate and stabilize the training of language-vision models, which is particularly important for large-scale language-vision applications.
2. The analysis of relationship between loss spike and AdamW second moment is interesting.

**Weaknesses:**

1. My main concern is about the novelty. The idea of using high-precision for backward weight gradients is studied by previous works such as the Gradients Bifurcation methd [1]. Float8 training are also widely studied. For the proposed StableAdamW, I didn't see the essential difference with AdaFactor, despite the explanation around line 255.
2. It seems that the proposed methods are not designed particularly for language-vision models. It is better to compare with existing methods on traditional CV or NLP tasks.

minor issues:

line 331, divides the learning rate for iteration $t$ by $max(RMS_t,1)$.

line 154, why constrains the float8 with in -1 and 1 ? float8cast(x/absmax(x))

[1] Scalable Methods for 8-bit Training of Neural Networks.

**Questions:**

See the weaknesses.

**Limitations:**

yes

---

> ### Author Rebuttal · Authors · 2023-08-07
>
> Thank you very much for your review, we hope this rebuttal addresses your concerns.
>
> **Novelty**
>
> We thank you for highlighting the gradient bifurcation method, and we will update the paper to include this important reference. As prior work has observed (e.g., [1]), quantization becomes more difficult at scale. Therefore, we believe that our experiments in this large scale setting, and for large vision-language models in particular, are valuable for the community. We emphasize that the largest vision models that have been studied before have been an order of magnitude smaller. In addition, we show that the technique can be simplified for float8 vs. int8.
>
> Moreover, we do not claim methodological novelty over AdaFactor, but instead aim to draw the community's attention to an important feature of AdaFactor that is often overlooked, and show that this feature can be effectively transferred to Adam. AdaFactor is decreasing in popularity as it has been shown to underperform Adam at scale [2]. However, our results indicate that this may be due to factored moments and not some of AdaFactor’s other innovations which can effectively be transferred to Adam. Finally, our work is novel as we establish a predictive relationship for loss spikes.
>
> **Why CLIP?**
>
> We focus on the contrastive vision-language pre-training setting because we believe that it is of increasing importance as it underlies approaches from generative models to zero-shot classifiers. While the results may be more general, contrastive approaches require a very large batch size. As we discuss in Section D.3 of the supplementary material (L686), we believe that this distinction explains why LLM.int8() underperforms the bfloat16 baseline in the setting that we consider.
>
> **Why constrain the float8 to [-1, 1]?**
>
> We apologize for our lack of clarity – this is a typo. After converting to [-1, 1] range we multiply by the max value in float8.
>
> [1] https://arxiv.org/abs/2208.07339
>
> [2] https://arxiv.org/abs/2112.11446

---

> > ### Comment · Reviewer_AbFj · 2023-08-14
> > **Response to rebuttal**
> >
> > Thank the authors for the response! My main concern is still the lack of novelty. The proposed SwitchBack is previously studied in Gradients Bifurcation, the proposed StableAdamW has no methodological novelty over AdaFactor as responsed by the authors. I understand the authors' statement about drawing the community's attention to the useful methods studied in this paper, however, I think the contribution is not enough to get accepted.
> >
> > I think it it critical for a training method to generalize well. Otherwise, as raised by Reviewer rd6a, 'if the training fails, it's unclear if the failure stems from precision issues or model issues'. Thus, I think it is critical to test the generalization ability of the proposed method to a wide range of tasks and models.
> >
> > I have also read other reviewers' comments, but I still tend to reject this paper based on the above two concerns, which are not well addressed by the authors' response.

---

> > > ### Author Response · Authors · 2023-08-15
> > >
> > > Thank you very much for engaging and for your feedback, while we respectfully disagree we sincerely appreciate your time. We do highlight that even despite the contributions for which you have novelty concerns, there are still two remaining contributions which we believe to be important: a) We demonstrate successful float8 training with quantized gradients, weights, and activations by using zero-init layer-scale (Figure 2). b) We establish a predictive relationship between RMS spikes in the patch embedding layer and loss spikes (Figure 4 and Figures 15-20 of the appendix). Moreover, with respect to the downstream task concern we note that [1] (e.g., Figure 8) and [2] have observed a strong correlation between zero-shot and downstream task performance.
> > >
> > > [1] https://arxiv.org/abs/2103.00020
> > >
> > > [2] https://arxiv.org/abs/2212.07143

---

### Decision · Program_Chairs · 2023-09-21

**Decision:**

Accept (poster)

**Comment:**

This paper develops a simple and effective strategy to enhance the training efficiency and stability of large vision-language models. Overall, all reviewers appreciate its clear presentation, strong motivation, and solid empirical results. A few concerns are raised, including 1) novelty seems not enough; 2) experiments are only on CLIP models; and 3) additional ablations/clarifications are needed.

The rebuttal addresses most of them, but reviewers are still concerned that the technique novelty of this paper is not very strong. While the ACs acknowledge that this concern is legitimate, they recognize the value of this work and believe the research community will be interested in this type of work on accelerating the training of large-scale vision-language models. Weighing the pros and cons, the ACs conclude that the paper's advantages surpass its limitations and thus recommend its acceptance.